# Inhibition of a nutritional endosymbiont by glyphosate abolishes mutualistic benefit on cuticle synthesis in *Oryzaephilus surinamensis*

Julian Simon Thilo Kiefer [1], Suvdanselengee Batsukh[1], Eugen Bauer[1], Bin Hirota[2,3], Benjamin Weiss [1,4], Jürgen C. Wierz [1], Takema Fukatsu [2,3,5], Martin Kaltenpoth [1,4,6] & Tobias Engl[1,4,6 ✉]

Glyphosate is widely used as a herbicide, but recent studies begin to reveal its detrimental side effects on animals by targeting the shikimate pathway of associated gut microorganisms. However, its impact on nutritional endosymbionts in insects remains poorly understood. Here, we sequenced the tiny, shikimate pathway encoding symbiont genome of the saw-toothed grain beetle *Oryzaephilus surinamensis*. Decreased titers of the aromatic amino acid tyrosine in symbiont-depleted beetles underscore the symbionts' ability to synthesize pre-phenate as the precursor for host tyrosine synthesis and its importance for cuticle sclerotization and melanization. Glyphosate exposure inhibited symbiont establishment during host development and abolished the mutualistic benefit on cuticle synthesis in adults, which could be partially rescued by dietary tyrosine supplementation. Furthermore, phylogenetic analyses indicate that the shikimate pathways of many nutritional endosymbionts likewise contain a glyphosate sensitive 5-enolpyruvylshikimate-3-phosphate synthase. These findings highlight the importance of symbiont-mediated tyrosine supplementation for cuticle biosynthesis in insects, but also paint an alarming scenario regarding the use of glyphosate in light of recent declines in insect populations.

[1] Evolutionary Ecology, Institute of Organismic and Molecular Evolution (iomE), Johannes Gutenberg University, Mainz, Germany. [2] Bioproduction Research Institute, National Institute of Advanced Industrial Science and Technology (AIST), Tsukuba, Japan. [3] Department of Biological Sciences, Graduate School of Science, University of Tokyo, Tokyo, Japan. [4] Research Group Insect Symbiosis, Max-Planck-Institute for Chemical Ecology, Jena, Germany. [5] Graduate School of Life and Environmental Sciences, University of Tsukuba, Tsukuba, Japan. [6] Department of Insect Symbiosis, Max-Planck-Institute for Chemical Ecology, Jena, Germany. ✉email: tengl@ice.mpg.de

Glyphosate is a widely used, broad-spectrum herbicide that targets the shikimate pathway by inhibition of the 5-enolpyruvylshikimate-3-phosphate synthase (EPSPS)[1,2]. The shikimate pathway is present in plants, fungi, and bacteria to biosynthesize the aromatic amino acids phenylalanine, tyrosine and tryptophan, as well as folates. As the inhibition of the EPSPS by glyphosate in susceptible plants is lethal[3], this herbicide is extensively used in agriculture in combination with genetically modified glyphosate-resistant crops to eliminate competing plants[4]. Animals, by contrast, are assumed to be unaffected by glyphosate, as they lack the shikimate pathway and meet their demands for aromatic amino acids from external sources[5].

However, animals do not live in isolation but engage in manifold mutualistic interactions with microorganisms[6]. While the microorganisms mostly gain the imminent advantage of a stable environment and the provision of basic nutrients[7], animals benefit from the more specialized metabolic capabilities of the symbiont, giving both of them a competitive advantage or the ability to expand to a certain niche[8–14]. Specifically, animals often benefit from the metabolic capabilities of microorganisms that they lack themselves, for example, the synthesis of essential amino acids and (B-)vitamins, which is especially important when the insect host feeds on nutritionally unbalanced diets like plant sap or blood[14,15]. In the same way, semi-essential nutrients, i.e. nutrients like the aromatic amino acid tyrosine that can in principle be derived from the other aromatic amino acids taken up from the diet, but are in certain life stages like the metamorphosis of holometabolic insects still limited, are supplied by many nutritional endosymbionts[16–19]. Despite several studies emphasizing that glyphosate displays only "minimal toxicity" in animals via off-target activity due to the lack of the shikimate pathway in their genomes[1,20,21], recent work demonstrates that glyphosate does have a negative impact on insects, either directly or by inhibiting the EPSPS of animal-associated, mutualistic bacteria, with negative fitness consequences for the host[22,23]. In blood-feeding tsetse flies, glyphosate interferes with the biosynthesis of folate by the γ-proteobacterial symbiont *Wigglesworthia glossinidia*[24]. The herbicide also alters the gut microbiota of honeybees that consequently become more susceptible to opportunistic pathogens[25,26]. Since many obligate insect endosymbionts encode the shikimate pathway and supply aromatic amino acids to their host[19,27,28], glyphosate could prove highly detrimental for diverse insect hosts, with potentially severe ecological implications given the widespread nature of these mutualistic interactions in insects[29–31]. However, the impact of glyphosate on insect–microbe associations remains poorly studied, particularly with respect to obligate nutritional endosymbionts in herbivorous insects.

The cuticle of insects is primarily composed of chitin fibrils enveloped in protein chains rich in aromatic amino acids[32–34]. Additionally, insects modify the native cuticle through the integration of phenolic compounds derived from tyrosine in two processes called melanization and sclerotization, which lead to a cuticle that is darker, harder, and less permeable to water[35,36]. However, in many herbivorous diets nitrogen in general, but also specifically tyrosine, as well as phenylalanine and tryptophan from which tyrosine could be derived by insects are limited[37,38]. Among the holometabolous insects, beetles (Coleoptera) are distinguished by an especially strong adult cuticle, including the fully hardened forewings, i.e. the elytrae[39]. In addition, holometabolous insects develop their full imaginal cuticle during the pupal stage when the insect is undergoing a metamorphosis as well as during the first days as an imago[36]. As insects are not able to feed during metamorphosis, they are reliant on stored nutrients acquired during their larval development for metamorphosis and cuticle formation. In the case of tyrosine, the amount an insect can store is limited, because the amino acid is toxic in high concentrations[40]. In this situation, harboring an endosymbiont that can synthesize tyrosine or its precursors chorismate or prephenate[5,41] in the moment of demand represents a strategy to cope with the storage problem[16,19,42,43].

We recently described a bacteriome-localized Bacteroidetes endosymbiont in the sawtoothed grain beetle *Oryzaephilus surinamensis* (Coleoptera, Silvanidae)[42,43], which is a worldwide distributed pest of cereals and other stored food[44]. In contrast to many other nutritional endosymbionts, the symbiont of *O. surinamensis* can be removed by treating the beetle with antibiotics or heat without interrupting the host life cycle[42,43]. The loss of a nutritional endosymbiont usually has severe effects on its host, e.g. by arresting development and reproduction and/or causing high mortality[19]. In contrast, experimentally symbiont-deprived (aposymbiotic) *O. surinamensis* beetles are viable, able to reproduce, and can be maintained in stable aposymbiotic populations under laboratory conditions, allowing to disentangle the effect of symbiont loss from direct antibiotic treatment[42]. In comparison to symbiont-containing control beetles, aposymbiotic individuals exhibited a 30% thinner and noticeably lighter cuticle, which resulted in a significant fitness decrease due to higher mortality, especially under desiccation stress in stored grain products, the natural habitat of the beetle[42].

In this work, we demonstrate (i) that the symbiont genome of *O. surinamensis* is extremely streamlined, providing precursors for cuticle synthesis via the shikimate pathway, (ii) that exposure to glyphosate compromises the symbiont establishment in *O. surinamensis* and induces the fitness-relevant cuticular defects of the host in a similar manner as the complete loss of the symbiont, (iii) that these phenotypic effects can be partially rescued by dietary tyrosine supplementation, and (iv) that the shikimate pathways of *O. surinamensis* as well as many other nutritional endosymbionts contain class I EPSPSs that are predicted to be glyphosate-sensitive. These results experimentally validate the functionality of the shikimate pathway in an obligate endosymbiont and more generally demonstrate the severe impact of glyphosate on organisms that are dependent on bacterial endosymbionts.

## Results

**Symbiont genome is extremely reduced and GC poor**. We sequenced the metagenome of *O. surinamensis* combining short and long-read technologies (Illumina and ONT) into a hybrid assembly to gain first insights into its symbiont's metabolic capabilities. As expected, we could detect the 16S rRNA sequence of a Bacteroidetes bacterium in the assembly that matched PCR-based Sanger sequences from previous studies[42,43]. In total, 13 contigs were extracted from the metagenomic assembly via taxonomic classification, GC content filtering as well as by manually searching for tRNAs and ribosomal proteins of Bacteroidetes bacteria (Fig. 1b). The longest was 74,813 bp and the shortest 6352 bp in length. Together they had a length of 307,680 bp with an average GC content of 16.2%. The draft genome encoded for 299 genes and had a coverage of 120× with short-read sequences and 61× with long-read sequences.

The phylogenetic reconstruction based on the conserved Clusters of Orthologous Group (COG) genes confirmed the placement of the endosymbiont of *O. surinamensis* in a group of insect-associated Bacteroidetes bacteria and specifically the close relationship to *Blattabacterium* spp. and *Sulcia muelleri* (*S. muelleri*) that was already previously reported based on a 16S rRNA gene phylogeny (Supplementary Fig. 1)[42]. The symbiont's genome encodes for 299 protein-coding sequences, 28 tRNAs and 51 ribosomal proteins (21 SSU and 30 LSU proteins). Based on a

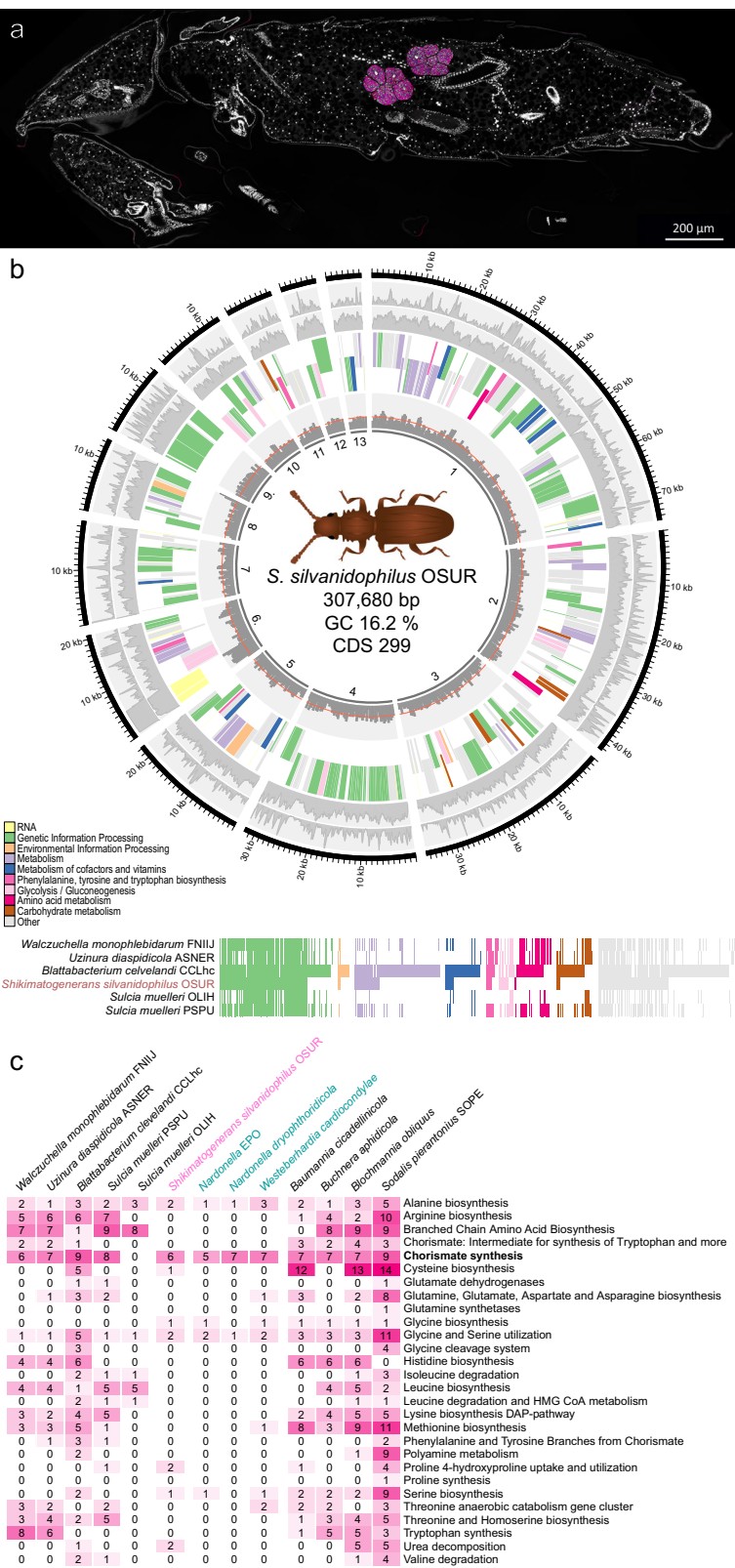

set of single-copy marker genes that are assumed to be essential, the *O. surinamensis* symbiont genome is estimated to be 66.7% complete[45]. However, this 'completeness' measure is based on essential genes in free-living bacteria and known to severely underestimate the completeness of highly eroded genomes of intracellular bacterial symbionts[46,47]. Concordantly, the estimation falls in the range of other insect-associated Bacteroidetes

mutualists that exhibit highly eroded, closed genomes (54.20–67.38% of *S. muelleri* PSPU and *W. monophlebidarum*; Supplementary Tables 1 and 2). Despite the remaining gaps in the symbiont genome sequence, the 13 contigs are thus inferred to contain the complete or almost complete set of coding sequences, which is corroborated by the presence of a complement of tRNAs, tRNA synthetases, and ribosomal proteins that is similar to other

**Fig. 1 *Shikimatogenerans silvanidophilus*, the symbiont of *O. surinamensis*. a** Fluorescence in situ hybridization micrograph of a sagittal section of a 5-day-old *O. surinamensis* pupa stained with CFB563mod-Cy3 (magenta) and DAPI (white). **b** Circular representation of the draft genome of *S. silvanidophilus*. Single contigs are sorted clockwise by length. The outer gray circles denote coverage with long- and short-reads, respectively, the intermediate circles indicate annotated functional KEGG categories separated by direction of transcription (see legend for depicted categories). The inner gray circle denotes relative GC content and the average GC content of 16.2% by the red line. Below: comparison of the functional gene repertoires of Bacteroidetes symbionts in insects. Box colors are based on KEGG's categories. **c** Detailed comparison of the amino acid metabolism gene repertoires between the *S. silvanidophilus* genome (pink), other Bacteroidetes symbionts (left) and Proteobacteria symbionts (right), some of which are known to exclusively provision tyrosine precursors to their insect host (blue).

Bacteroidetes endosymbionts. The contigs are likely also fragments of a single chromosome, as it does not feature duplicated genes, multiple rRNA operons or variable coverage across the different contigs, which are indicative of such fragmentation reported from multiple fragmented chromosomes of *Hodgkinia* strains of different *Magicicada* species[48–50]. As an extremely low GC content (16.2%) and long repeat sequences towards contig ends are known issues for sequencing technologies and PCRs[51–54], these features likely prevented successful amplification steps and the in silico assembly of the contigs into a single genome despite the high coverage (120× with short-read sequences and 61× with long-read sequences), especially as the GC content at the ends of most contigs dropped to even lower values (~4% within the last 100 bp).

We also compared the presence and arrangement of the genes between Bacteroidetes endosymbionts of different host insects. A synteny plot (Supplementary Fig. 2) revealed that there is no conserved arrangement of genes between the genomes of related Bacteroidetes symbionts. This likely explains futile attempts to assemble the genome by mapping the assembled contigs to *Blattabacterium* or *S. muelleri* as reference genomes or to sort contigs into a single scaffold.

### *Candidatus* Shikimatogenerans silvanidophilus OSUR encodes glycolysis and shikimate pathways.

The metabolic repertoire of the *O. surinamensis* symbiont is highly reduced (Fig. 1b). Apart from general genetic information processing including DNA replication and repair, transcription and translation, it only encodes an extremely limited set of metabolic pathways including a full glycolysis pathway to process glucose-6-phosphate to erythrose 4-phosphate (E4P) and phosphoenolpyruvate (PEP). In addition the genome encodes all the genes of the shikimate pathway except a shikimate dehydrogenase (*aroE* [EC:1.1.1.25]) that utilizes PEP and E4P to produce chorismate, as well as a chorismate mutase to catalyze the conversion of chorismate into prephenate, the precursor of the aromatic amino acids phenylalanine, tryptophan, and tyrosine (Fig. 2). The lack of *aroE* is described in other tyrosine-supplementing bacterial symbionts: *Cd* Carsonella ruddii in *Pachypsylla venusta*[55] and *Cd* Nardonella EPO in *Euscepes postfasciatus*[19]. AroD (3-dehydroquinate dehydratase [EC:4.2.1.10]) and *aroE* are also often found as a bifunctional enzyme *aroDE* (3-dehydroquinate dehydratase/shikimate dehydrogenase [EC:4.2.1.10 1.1.1.25])[56], possibly complementing the loss of one of the enzymes. Like all other Bacteroidetes symbionts, the *O. surinamensis* symbiont genome encodes the bifunctional *aroG/pheA* gene (phospho-2-dehydro-3-deoxyheptonate aldolase/chorismate mutase [EC:2.5.1.54 5.4.99.5]). The genomic data revealed no transporter for glucose, so it remains unknown how the symbiont acquires the substrate for glycolysis from the host (Fig. 2).

Whether the *O. surinamensis* symbiont can recycle nitrogen, like *Blattabacterium* sp.[57,58] remains unclear, as it encodes the urease α and γ subunits (*ureC* [EC:3.5.1.5]), but no glutamate dehydrogenase (*gdhA*) that would allow integrating the resulting ammonium into the amino acid metabolism via glutamate was

detected. Instead, it encodes a proline transporter (*opuE*), proline dehydrogenase [EC:1.5.5.2], and oxidoreductase [EC:1.2.1.88] to import and convert proline into glutamate[59,60], which may then be exported to function as an amino group donor for the synthesis of the aromatic amino acids from chorismate/prephenate by the beetle itself[19]. This alternative pathway for glutamate synthesis was not described in *Blattabacterium*[61], *S. muelleri*[62], *Uzinura diaspidicola*[63], or *Walczuchella monophlebidarum*[64].

A comparison of the metabolic gene repertoires for amino acid biosynthesis revealed convergent genome erosion between the *Oryzaephilus* symbiont and ɣ-proteobacterial symbionts known to provision precursors for the host's cuticle biosynthesis (Fig. 1c). We observed a gradual loss of functions across the Bacteroidetes insect symbionts, with the *O. surinamensis* symbiont genome exhibiting the most strongly reduced repertoire of biosynthetic genes. A convergent reduction of metabolic functions was observed in the ɣ-proteobacterial symbionts *Westerberhardia* and *Nardonella* sp., whose sole metabolic function appears to be the provisioning of aromatic amino acid precursors that are in high demand for cuticle biosynthesis of their insect host[19,27,65].

Based on our findings, we propose the name '*Candidatus* Shikimatogenerans silvanidophilus OSUR' for this endosymbiont of *O. surinamensis*, henceforth called *S. silvanidophilus*. The genus name *Shikimatogenerans* refers to its ability to perform the shikimate pathway. Previous studies have shown that there might be other closely related Bacteroidetes bacteria associated with other beetle families[42,66,67]. Thus, we propose *silvanidophilus* as species name to indicate that this symbiont is associated with beetles of the family Silvanidae. As the same studies also revealed that *O. mercator* has a similar symbiont we also propose to add OSUR to identify the strain associated with *O. surinamensis*.

### Amino acid titers are influenced by symbiont presence.

Using the genomic data as a basis, we then asked whether the symbiont-encoded pathways are indeed functional. To this end, we tested if the symbiont presence indeed influenced the titer of aromatic amino acids. This would be expected when the symbiont is delivering prephenate to the host, a precursor that can also be transformed to tyrosine or phenylalanine by the beetles themselves[17,27,68]. Due to the presence of the proline transporter and proline converting enzymes in the symbiont genome, an influence on proline titer and glutamic acid was also expected based on the metabolic reconstruction. After hatching from the egg, the larva of *O. surinamensis* spends weeks in multiple instars before it pupates and hatches 5 days later as an imago. During metamorphosis, the biosynthesis of the adult cuticle starts and continues in the first days of the imago[36]. As tyrosine is used in copious amounts in the cuticle biosynthesis[69] amino acid titers might change dynamically. Thus, we expected the major symbiont contribution of tyrosine in the pupal stage and early adulthood[19]. Accordingly, we found an overall significantly positive influence of the symbiont presence on the titer of tyrosine (FDR-corrected GLMs, $p = 0.000086$, see Fig. 3 and Supplementary Fig. 3 and detailed statistic results for all the following values in Supplementary data 1 and Table 1). Specifically, late

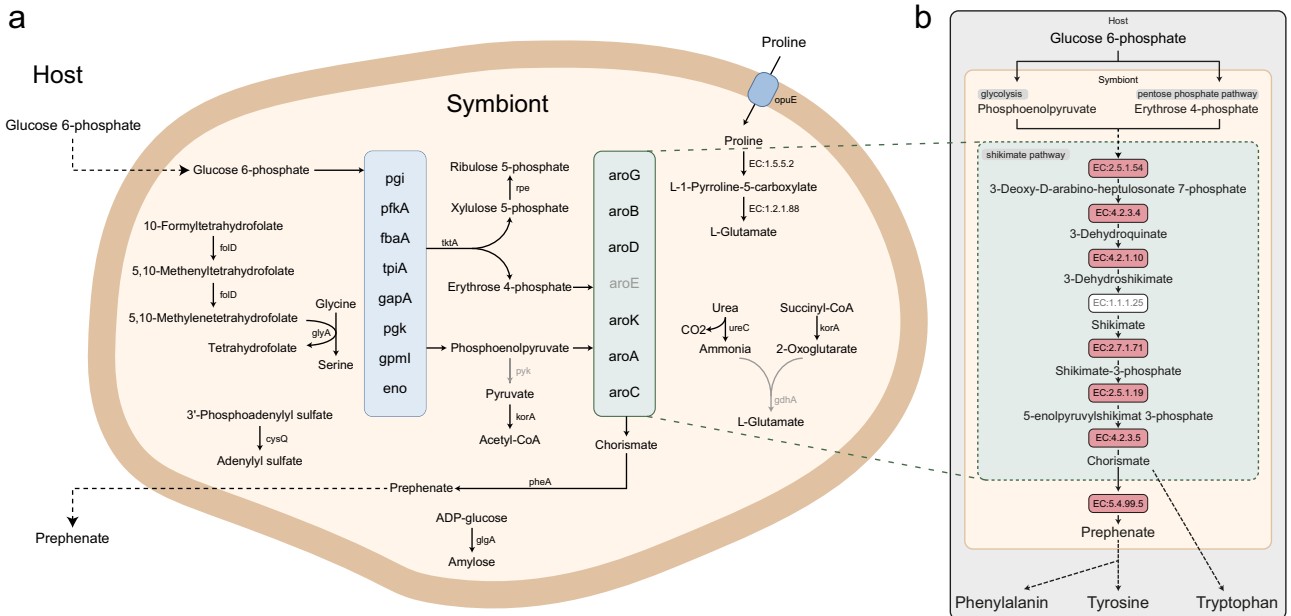

**Fig. 2 Metabolism of the symbiont *S. silvanidophilus*. a** Complete, reconstructed metabolism of *S. silvanidophilus*, as inferred from genomic data. Enzymes and arrows in gray were missing in the genome annotation. Dashed arrows indicate transport processes without annotated transporters, but which are expected to occur based on observed phenotypes. **b** Schematic diagram of the shikimate pathway in *S. silvanidophilus*. Dashed arrows represent multiple enzymatic steps. Red boxes: enzyme present in the genome. White box: enzymes not annotated in the genome, but reaction probably catalyzed by another enzyme.

symbiotic pupae had a significantly higher tyrosine titer than aposymbiotic ones (Wilcoxon-rank-sum tests with FDR-corrected $p$-values: $p = 0.00078$), shortly before adult emergence. By contrast, there was a negative impact of symbiont presence on proline and glutamic acid levels (FDR-corrected GLMs, $p = 0.00012$ and $p = 0.000057$), which are interconverted by the symbiont based on the genomic prediction and used to synthesize tyrosine from prephenate. Again, free proline was significantly lower in late symbiotic vs. aposymbiotic pupae (FDR-corrected Wilcoxon-rank-sum test, $p = 0.00078$) and bound proline in early adults (FDR-corrected Wilcoxon-rank-sum test, $p = 0.00078$). Free glutamic acid was significantly lower in early and late symbiotic pupae (FDR-corrected Wilcoxon-rank-sum tests, both $p = 0.00078$, Fig. 3) and the elytrae of early symbiotic adults (FDR-corrected Wilcoxon-rank-sum test, $p = 0.0059$). There was no difference in the titer of bound or cuticular tyrosine in the different life stages of the beetle in direct comparison (FDR-corrected Wilcoxon-rank-sum test, $p > 0.05$) and only protein-bound glutamic acid of the full body of early symbiotic beetles was significantly higher than in aposymbiotic ones (FDR-corrected Wilcoxon-rank-sum test, $p = 0.0059$). All other amino acids were not influenced by symbiont presence alone, although we detected several interaction effects (FDR-corrected GLMs, $p > 0.005$; Supplementary data 1).

**The symbiont's shikimate pathway is sensitive to glyphosate and its inhibition results in an aposymbiotic phenotype**. Next, we investigated the consequences of a pharmacological inhibition of the shikimate pathway on the symbiosis. Therefore, we made use of the pesticide glyphosate, an allosteric inhibitor of the enzyme EPSPS (5-enolpyruvoylshikimate-3-phosphate synthase) which is encoded by *aroA*[1]. We experimentally determined the impact of glyphosate exposure and aromatic amino acid supplementation during the entire larval and early adult development on phenotypic parameters that were previously demonstrated to be impacted by the symbiosis, i.e. cuticle thickness and

melanization[42,43]. As predicted, both the thickness (Kruskal–Wallis $\chi^2 = 27.3$, df = 7, $p$-value = 0.0001; Fig. 4a, Table 3) and melanization (Kruskal–Wallis $\chi^2 = 36.0$, df = 7, $p$-value = 0.0001; Fig. 4b, Table 3) of the cuticle were influenced by glyphosate exposure and by aromatic amino acid supplementation of the beetle diet. Symbiotic beetles showed a significant reduction of both cuticle traits after inhibition of the shikimate pathway with 1% glyphosate (Dunn's test: symbiotic vs. symbiotic + glyphosate: $p = 0.0047$ (thickness) and $p = 0.0049$ (melanization)), to the same level as observed in aposymbiotic beetles (Dunn's test: aposymbiotic vs. symbiotic: $p = 0.0005$ (thickness) and $p = 0.0008$ (melanization); aposymbiotic vs. symbiotic + glyphosate: $p = 0.73$ (thickness) and $p = 0.54$ (melanization)). A lower amount of glyphosate (0.1%) led to intermediate phenotypes that did not differ significantly from either symbiotic or aposymbiotic or 1% glyphosate treated individuals (Dunn's test: $p > 0.05$). Importantly, the thinner and less melanized cuticle of both aposymbiotic and glyphosate-exposed (1%) symbiotic beetles could be rescued to an intermediate state by adding aromatic amino acids to the diet, resulting in cuticular traits that differed neither from symbiotic nor aposymbiotic or 1% glyphosate-treated beetles (Dunn's test: $p > 0.05$). By contrast, aromatic amino acid supplementation did not affect cuticular traits of symbiotic beetles that were not exposed to glyphosate (Dunn's test: $p > 0.05$). These findings support the genome-based prediction that *S. silvanidophilus* supplements precursors for tyrosine biosynthesis via the shikimate pathway.

The two plausible scenarios for the impact of glyphosate exposure on symbiont contributions are (i) a reduced chorismate biosynthesis by a stable symbiont population, or (ii) an overall decrease in symbiont titers due to glyphosate exposure. This includes either direct effects trough glyphosate toxicity, but also indirect effects via consequences of an inhibited chorismate biosynthesis like a lack of aromatic amino acids or host feedback mechanisms. Concordant with the latter hypothesis, we detected significant differences in symbiont titers of one-week-old symbiotic adults across experimental treatments (Kruskal–Wallis

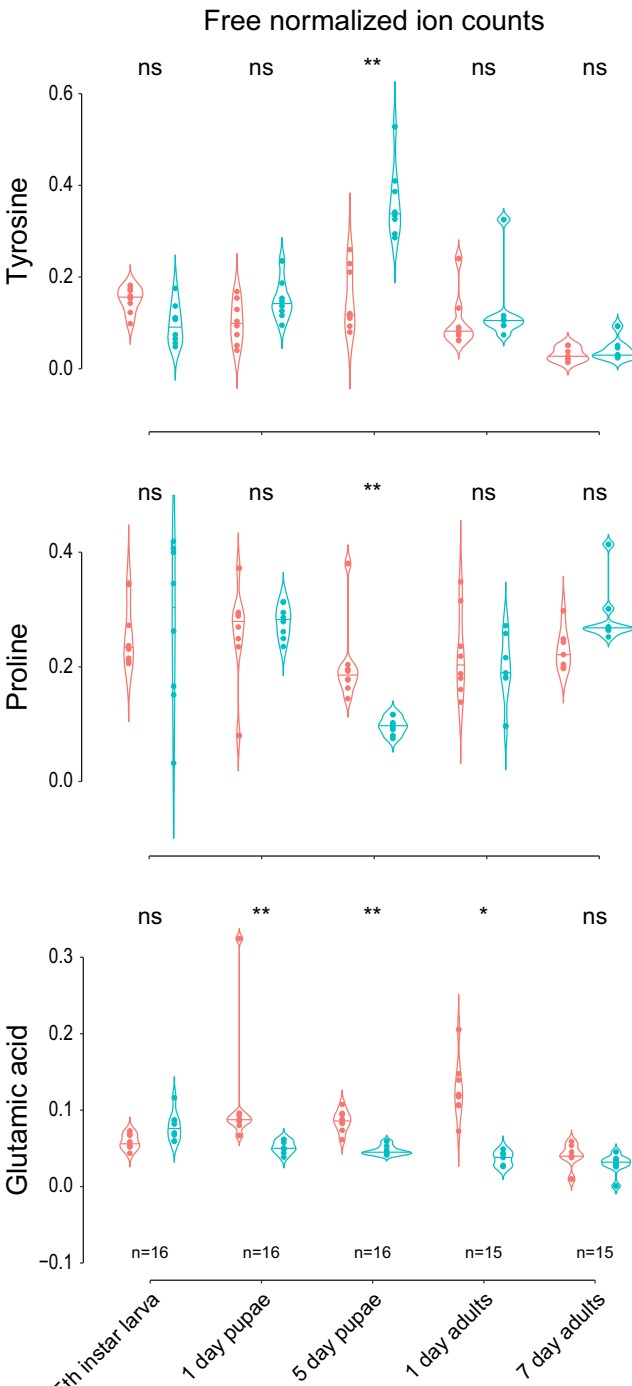

**Fig. 3 Comparison of titers of the three amino acids tyrosine, proline, and glutamic acid that were influenced by symbiont presence.** Shown are free amino acid titers in the whole body (without elytrae in case of adults) of symbiotic and aposymbiotic *O. surinamensis* beetles. Red: aposymbiotic beetles, Blue: symbiotic beetles. The data distribution is visualized with violin plots and an additional horizontal line depicting the median. The FDR-corrected unpaired, two-sided Wilcoxon-rank-sum-tests: ns $p > 0.05$, *$0.05 < p < 0.01$, **$p < 0.01$.

$\chi^2 = 59.0$, df = 7, $p$-value < 0.0001; Fig. 4c, Table 1). The symbiont titers in symbiotic beetles exposed to high glyphosate concentrations were significantly reduced by 98% (based on medians; Dunn's test: sym vs. 1% glyphosate: $p = 0.00034$). Low glyphosate concentrations and supplementation with aromatic amino acids also reduced symbiont titers to an intermediate level

that did not differ significantly from either untreated beetles nor those treated with high glyphosate amounts (Dunn's test: $p > 0.05$). In combination with low amounts of glyphosate, supplementation of aromatic amino acid reduced the symbiont titers further, leading to a significant difference to untreated beetles (Dunn's test: AA + 0.1% glyphosate vs untreated: $p = 0.028$).

Finally, to assess how common symbiont-mediated sensitivity to glyphosate is among insects engaging in nutritional symbioses, we tested whether the EPSPS of other intracellular symbionts besides the one of *S. silvanidophilus* belong to glyphosate sensitive or insensitive EPSPS variants by a phylogenetic analysis of their amino acid sequences[25]. The EPSPSs of all included Bacteroidetes and also Proteobacteria symbionts of insects clustered with class I EPSPS and are thus predicted to be sensitive to glyphosate—in contrast to insensitive class II enzymes of several free-living bacteria[70] (Fig. 5 and Supplementary Fig. 4).

## Discussion

The sawtoothed grain beetle *O. surinamensis* engages in an intimate association with the Bacteroidetes endosymbiont *S. silvanidophilus* that confers desiccation resistance via enhanced cuticle synthesis[42,43]. This physiological contribution represents a significant fitness benefit in dry habitats like mass grain storage[42]. Here, we demonstrated that the genome of *S. silvanidophilus* experienced a drastic erosion, resulting in a genome of 308 kbp in size and a strongly AT-biased nucleotide composition (16.2% GC), akin to what has been described for other obligate intracellular[47,71,72] or extracellular insect symbionts[73–76].

Moreover, the symbiont genome encodes only a few metabolic pathways, of which the largest remaining ones are the glycolysis and shikimate pathways. Phosphoenolpyruvate (PEP) and Erythrose-4-phosphate (E4P)—both originating from glycolysis—are used as substrates in the shikimate pathway to synthesize chorismate, which in turn is transformed to prephenate by the symbiont and then converted to the aromatic amino acids phenylalanine and tyrosine by the host, as recently demonstrated in the weevil-*Nardonella* endosymbiotic system[19]. Of the seven genes in the shikimate pathway (*aroG*, *aroB*, *aroD*, *aroE*, *aroK*, *aroA*, *aroC*), the genome only lacks the gene for shikimate dehydrogenase (*aroE*), which catalyzes the reversible NADPH-linked reduction of 3-dehydroshikimate to shikimate. However, the shikimate pathways of *Nardonella* EPO, the endosymbiont of the sweetpotato weevil *Euscepes postfasciatus* (Curculionidae: Cryptorhynchinae) and *Carsonella ruddii*, the endosymbiont of the gall-forming psyllid *Pachypsylla venusta* (Aphalaridae: Pachypsyllinae) also lack *aroE*, but remain functional[19,55], suggesting that the function can be taken over by other enzymes, either from the host or the endosymbiont—that have yet to be identified.

The genome also encodes for a urease (*ureC*, Urease α and γ subunits [EC:3.5.1.5]), so *S. silvanidophilus* may be able to recycle nitrogenous waste products of its host, as has been described for *Blattabacterium* in cockroaches and *Blochmannia* in carpenter ants[61,64,77]. Alternatively, however, this enzyme could be a non-functional remnant in the eroded genome, as no glutamate dehydrogenase (*gdhA*) synthesizing glutamate from ammonium and 2-oxogluterate was detected and the symbiont genome encodes no other annotated gene that could incorporate the resulting ammonium. Instead, the genome encodes for a proline transporter (*opuE*) and the genes to convert proline into glutamic acid (EC:1.5.5.2 and EC:1.2.1.88)[78,79]. Thus, proline as one of the most abundant amino acids in insect hemolymph might be utilized by the symbiont, among others as a source of glutamate[80]. As the genome encodes no other pathways of potential relevance

**Table 1 Results of pairwise Wilcoxon-rank-sum-tests from the R package 'ggpubr' for symbiont influence on glutamic acid, proline, and tyrosine titers in different life stages and different amino acid localizations.**

| Amino acid | Life stage | | Z | r | Lower 95% | Δ median | Upper 95% | p | Adjusted p |
|---|---|---|---|---|---|---|---|---|---|
| Tyrosine | Free | 5th instar larva | 2.26 | 0.56 | 0.01 | 0.06 | 0.10 | 2.07E−02 | 2.48E−01 |
| | | 1 day pupae | 1.63 | 0.41 | −0.10 | −0.05 | 0.01 | 1.05E−01 | 1.00E+00 |
| | | **5 day pupae** | **3.31** | **0.83** | **−0.29** | **−0.21** | **−0.12** | **1.55E−04** | **1.86E−03** |
| | | 1 day adults | 1.45 | 0.37 | −0.05 | −0.02 | 0.02 | 1.52E−01 | 1.00E+00 |
| | | 7 day adults | 0.75 | 0.19 | −0.03 | −0.01 | 0.01 | 4.63E−01 | 1.00E+00 |
| | Bound | 5th instar larva | 0.41 | 0.10 | −0.05 | −0.01 | 0.05 | 6.94E−01 | 1.00E+00 |
| | | 1 day pupae | 0.66 | 0.20 | −0.13 | 0.02 | 0.06 | 5.27E−01 | 1.00E+00 |
| | | 5 day pupae | 1.59 | 0.40 | −0.10 | −0.02 | 0.02 | 1.14E−01 | 1.00E+00 |
| | | 1 day adults | 1.73 | 0.43 | −0.11 | −0.07 | 0.01 | 8.30E−02 | 9.96E−01 |
| | | 7 day adults | 0.37 | 0.09 | −0.06 | −0.02 | 0.03 | 7.21E−01 | 1.00E+00 |
| | Elytrae | 1 day adults | 2.47 | 0.62 | −0.11 | −0.06 | −0.02 | 1.04E−02 | 1.25E−01 |
| | | 7 day adults | 2.26 | 0.56 | −0.04 | −0.02 | −0.01 | 2.07E−02 | 2.48E−01 |
| Proline | Free | 5th instar larva | 0.37 | 0.09 | −0.17 | −0.05 | 0.11 | 7.21E−01 | 1..00E+00 |
| | | 1 day pupae | 0.37 | 0.09 | −0.05 | −0.01 | 0.04 | 7.21E−01 | 1..00E+00 |
| | | **5 day pupae** | **3.31** | **0.83** | **0.06** | **0.09** | **0.11** | **1.55E−04** | **1..86E−03** |
| | | 1 day adults | 0.17 | 0.04 | −0.05 | 0.01 | 0.12 | 8.67E−01 | 1.00E+00 |
| | | 7 day adults | 2.49 | 0.64 | −0.08 | −0.05 | −0.02 | 9.32E−03 | 1.12E−01 |
| | Bound | 5th instar larva | 2.14 | 0.55 | 0.00 | 0.02 | 0.03 | 2.89E−02 | 3.47E−01 |
| | | 1 day pupae | 2.36 | 0.71 | 0.00 | 0.02 | 0.03 | 1.21E−02 | 1.45E−01 |
| | | 5 day pupae | 2.01 | 0.50 | 0.00 | 0.02 | 0.04 | 4.18E−02 | 5.02E−01 |
| | | **1 day adults** | **3.31** | **0.83** | **0.02** | **0.03** | **0.04** | **1.55E−04** | **1.86E−03** |
| | | 7 day adults | 0.68 | 0.17 | −0.01 | 0.00 | 0.01 | 5.05E−01 | 1.00E+00 |
| | Elytrae | 1 day adults | 2.05 | 0.51 | 0.01 | 0.03 | 0.05 | 3.79E−02 | 4.55E−01 |
| | | 7 day adults | 1.73 | 0.43 | 0.00 | 0.01 | 0.02 | 8.30E−02 | 9.96E−01 |
| Glutamic acid | Free | 5th instar larva | 2.36 | 0.59 | −0.03 | 0.02 | 0.00 | 1.48E−02 | 1.78E−01 |
| | | **1 day pupae** | **3.31** | **0.83** | **0.03** | **0.04** | **0.05** | **1.55E−04** | **1.86E-03** |
| | | **5 day pupae** | **3.31** | **0.83** | **0.03** | **0.04** | **0.05** | **1.55E−04** | **1.86E−03** |
| | | **1 day adults** | **3.18** | **0.82** | **0.06** | **0.08** | **0.11** | **3.11E−03** | **3.73E−02** |
| | | 7 day adults | 1.91 | 0.49 | 0.00 | 0.01 | 0.03 | 5.41E−02 | 6.49E−01 |
| | Bound | 5th instar larva | 1.79 | 0.46 | −0.03 | −0.02 | 0.00 | 7.21E−02 | 8.65E−01 |
| | | 1 day pupae | 1.42 | 0.43 | −0.04 | −0.01 | 0.01 | 1.64E−01 | 1.00E+00 |
| | | 5 day pupae | 2.33 | 0.58 | −0.03 | −0.01 | 0.00 | 1.64E−02 | 1.97E−01 |
| | | 1 day adults | 2.99 | 0.75 | −0.03 | −0.02 | −0.01 | 1.09E−03 | 1.31E−02 |
| | | 7 day adults | 0.79 | 0.20 | −0.02 | −0.01 | 0.04 | 4.42E−01 | 1.00E+00 |
| | Elytrae | **1 day adults** | **2.78** | **0.70** | **0.03** | **0.06** | **0.10** | **2.95E−03** | **3.54E−02** |
| | | 7 day adults | 0.00 | 0.00 | −0.02 | 0.00 | 0.02 | 1.00E+00 | 1.00E+00 |

Significant results (adjusted $p < 0.05$) are highlighted in bold. Z-test statistic, effect size r, lower and upper boundary of 95% confidence intervals and difference of medians, unmodified and Benjamini−Hochberg corrected p-values are reported. Significant results (adjusted $p < 0.05$) are highlighted in bold.

to the host, we hypothesized that the symbiont's beneficial impact on host fitness results from the supplementation of the tyrosine precursor prephenate, and possibly, from additional nitrogen recycling.

The quantification of amino acid titers throughout insect development in symbiotic and aposymbiotic beetles supports the genome-based predictions of symbiont-mediated tyrosine precursor biosynthesis and proline consumption. Both tyrosine and proline concentrations revealed significant differences between symbiotic and aposymbiotic beetles in late pupae, consistent with proline consumption and tyrosine biosynthesis during metamorphosis, which coincides with the reported maximum symbiont titers in *O. surinamensis*[81]. While our results indicate that the symbionts consume proline, the dynamics of proline conversion to glutamic acid remain elusive, as glutamic acid titers were lower in symbiotic than in aposymbiotic beetles across multiple life stages. Possibly, glutamic acid is predominantly utilized by the symbiont itself. Alternatively, as more chorismate is available for tyrosine synthesis in symbiotic than in aposymbiotic beetles, the glutamic acid titers might be constantly depleted, while tyrosine may only accumulate during metamorphosis, and is directly channeled into cuticle biosynthesis in all other life stages, as tyrosine accumulation may be toxic[40]. A

similar phenomenon of increased symbiont titers and intense cuticle synthesis and modification during metamorphosis and early adulthood was described in other pest beetles, the weevils *S. oryzae*[16] and *P. infernalis*[19], as well as the ant *Cardiocondyla obscurior*[27].

Symbiont-mediated contributions to cuticle biosynthesis evolved multiple times convergently in insects, with genomic evidence for tyrosine supplementation by symbionts in the ant genus *Cardiocondyla*, and both genomic and experimental support in carpenter ants[82,83], as well as the weevils *P. infernalis*[19] and *S. oryzae*[16,84]. The increasing number of tyrosine-supplementing symbioses provide evidence for this aromatic amino acid being a key nutrient that is limiting for many insects to produce their strongly sclerotized and melanized exoskeleton for protection against dessication and natural enemies[16,18,19,82]. While all the previously described symbioses that exclusively provision tyrosine precursors to their hosts involve γ-proteobacterial symbionts, *S. silvanidophilus* belongs to the Bacteroidetes, providing an example for functional convergence in genome-eroded symbionts across different bacterial phyla. Such convergence has been previously described for obligate endosymbionts of plant sap-sucking Hemiptera and Coleoptera that provision essential amino acids and/or vitamins to their

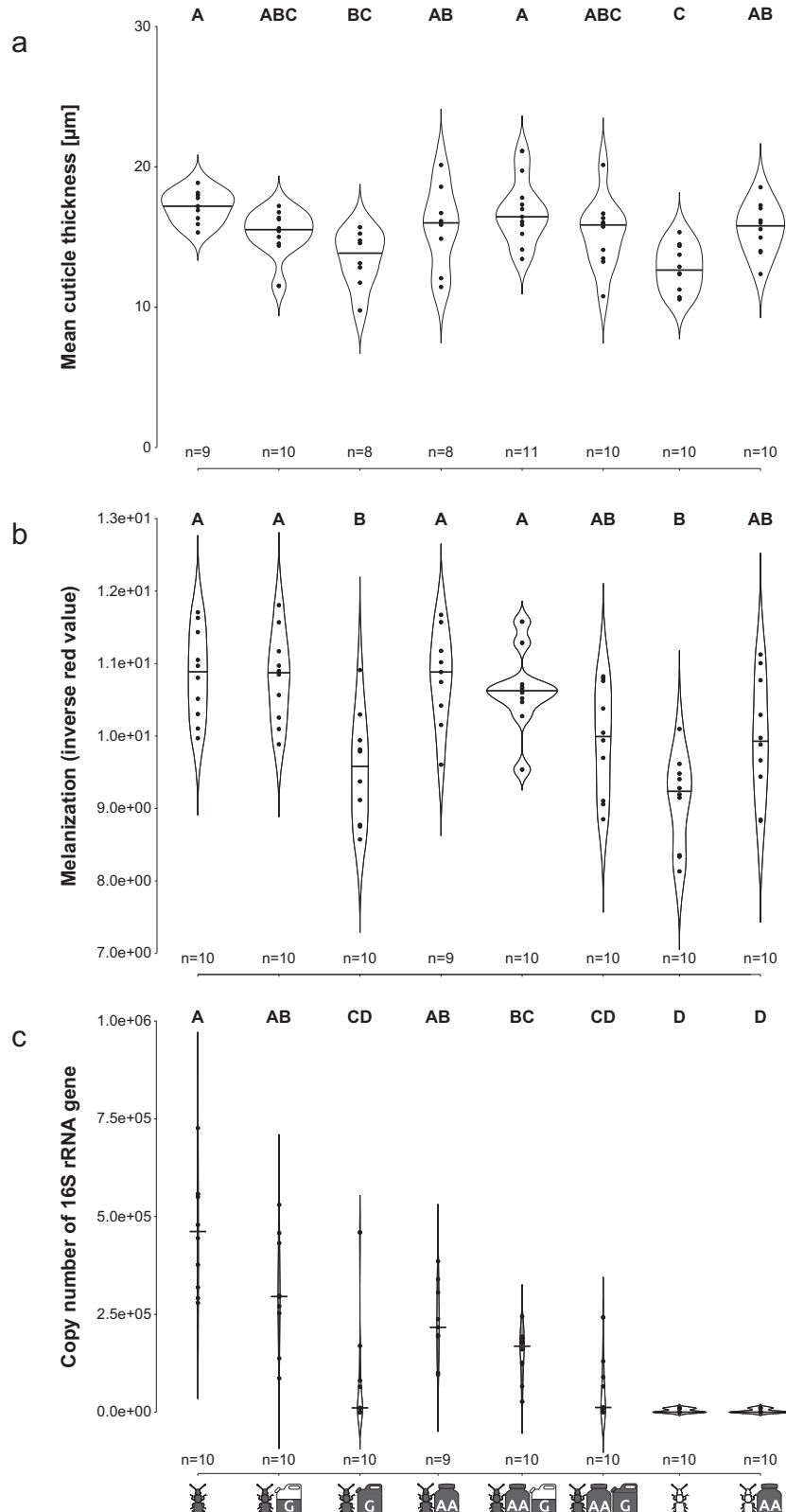

**Fig. 4 Effect of glyphosate exposure and aromatic amino acid supplementation on cuticle traits and symbiont titers in symbiotic and aposymbiotic beetles.** Cuticle thickness (**a**), melanization measured as thorax coloration (**b**), and symbiont titers (**c**) of aposymbiotic and symbiotic adults reared on different food compositions. See Table 2 for explanation of the symbols. The data distribution is visualized with violin plots and an additional horizontal line depicting the median. Different letters indicate significant differences between experimental treatments (Dunn's Test, α ≤ 0.05).

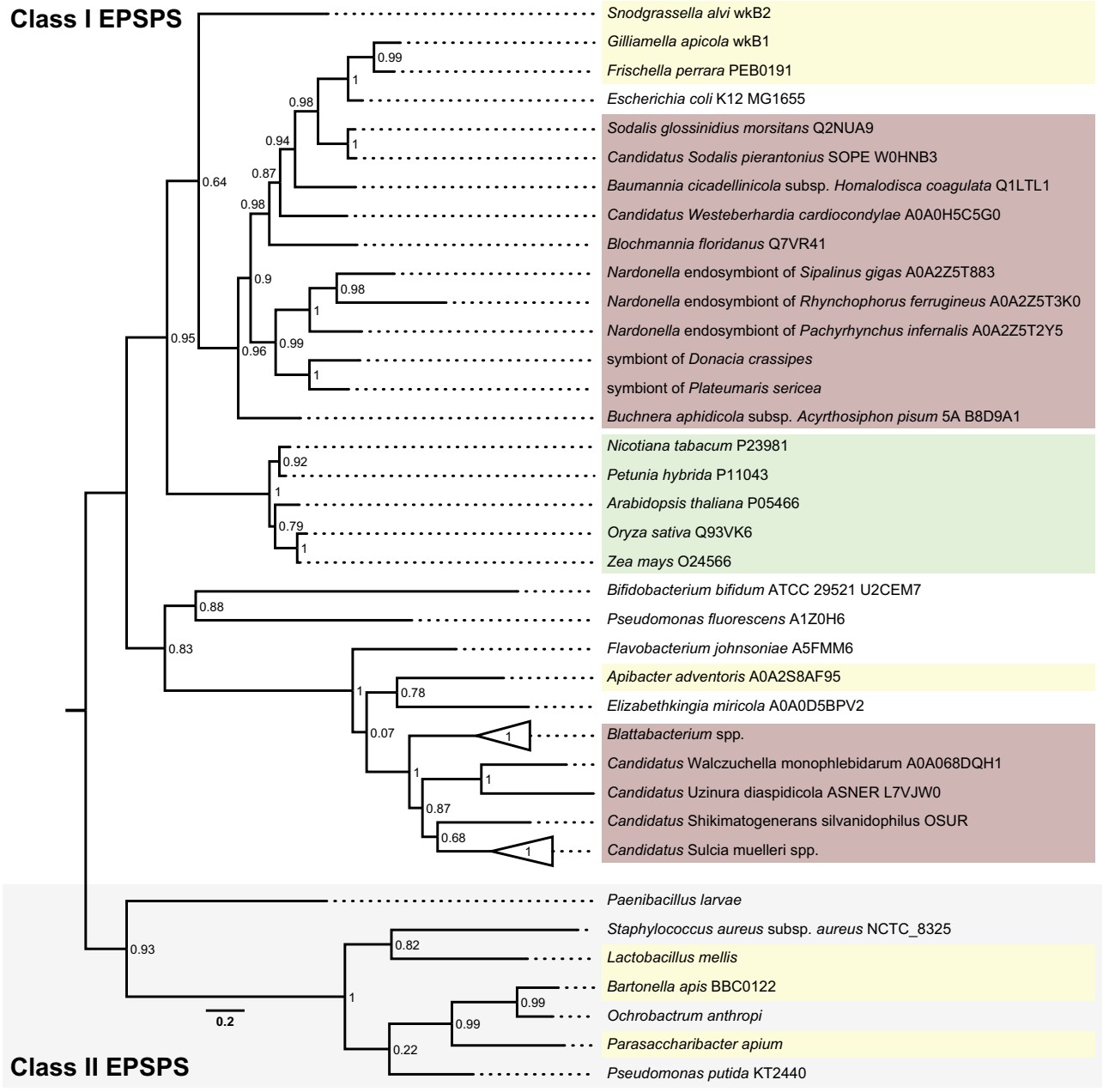

**Fig. 5 Phylogenetic classification of EPSPS enzymes.** Enzymes from different Bacteroidetes and γ-Proteobacteria insect symbionts as well as some free living bacteria and plants were classified based on FastTree and maximum- likelihood analyses of amino acid sequences of EPSPSs using the Jones–Taylor–Thorton model. Enzymes of plants (green), bacterial symbionts in the honeybee gut (yellow), and obligate intracellular insect symbionts (red) are highlighted. Node values indicate FastTree support values.

hosts[62,85,86]. Interestingly, the Bacteroidetes clade containing *S. silvanidophilus* comprises exclusively insect-associated bacteria, suggesting the possibility that the ancestor was a successful symbiont or pathogen of insects, reminiscent of the widespread reproductive manipulator *Wolbachia* and the common insect associate *Sodalis*[72,87,88].

Symbioses relying on nutritional supplements derived from the shikimate pathway are prone to inhibition of the *aroA*-encoded EPSPS by the herbicide glyphosate. Glyphosate sensitivity has already been used as an experimental tool to manipulate the obligate symbiont of tsetse flies, *Wigglesworthia morsitans*: chorismate-derived folate (Vitamin B9) biosynthesis by *Wigglesworthia* was inhibited by glyphosate, resulting in delayed larval development[24]. Furthermore, glyphosate exposure was found to

have a detrimental effect on the honey bee gut microbiota, which translated into increased mortality of the honeybees under pathogen pressure[1,25,26,70,89]. Concordantly, our results on an intracellular beetle symbiosis show that exposure to agronomically applied[90] or previously tested glyphosate levels[24,25] decreases symbiont titers and recapitulates cuticular phenotypes of aposymbiotic beetles, indicating that the inhibition of the symbiont's shikimate pathway results in aromatic amino acid starvation of both host and symbiont. Aromatic amino acid supplementation to glyphosate-exposed beetles partially rescued the host phenotype, but not the symbiont titers, indicating that the host uses the dietary tyrosine preferentially or exclusively for its own supply.

Interestingly, aromatic amino acid supplementation to symbiotic beetles suppressed the establishment of the symbiont

population in the host to a similar level as glyphosate, suggesting that either the symbionts cannot grow in the absence of self-produced precursors for aromatic amino acids or the host may sanction its symbionts when the latter's nutrient supplementation is no longer needed. While this phenomenon has been documented in some dynamic partnerships of plants and their root-associated mycorrhizae and rhizobia[91,92], it seems surprising in an intimate and co-evolved mutualism that may have been expected to show little potential for conflict between the partners[72]. Conceivably, however, symbiont titers may be regulated by the host based on tyrosine or L-DOPA concentrations during normal host development, resulting in symbiont suppression in times of high tyrosine availability.

The use of glyphosate in agriculture is currently heavily debated, based on increasing evidence for its detrimental impact on animals due to symbiont depletion[25,26,89,93] or inhibition of cuticle melanization[22]. Our findings on the glyphosate susceptibility of a beetle via its prephenate-supplementing endosymbiont exacerbate these concerns, particularly when considering our predictions based on an EPSPS phylogeny that many insects harbor endosymbionts susceptible to glyphosate. Whether insects whose symbionts provision tyrosine or its precursors among other amino acids or vitamins[28,65,74,94] equally suffer from glyphosate exposure needs to be experimentally tested and will sharpen our predictions. However, the widespread occurrence of nutritional endosymbionts relying on shikimate pathway-derived nutrients paints an alarming picture and suggests that glyphosate application holds a tremendous risk of severe ecological impacts. Especially in light of recent declines in the number and diversity of insects[56,95,96] and its impact on higher trophic levels[97–101], the use of herbicides with potential side-effects on animals or their associated microorganisms should be carefully reconsidered.

## Methods

**Insect cultures**. The initial *Oryzaephilus surinamensis* culture (strain JKI) was obtained from the Julius-Kühn-Institute/Federal Research Center for Cultivated Plants (Berlin, Germany) in 2014 and kept in culture since then. Continuous symbiotic and aposymbiotic (see below) *O. surinamensis* cultures were maintained in 1.8-L plastic containers, filled with 50 g oat flakes, at 28 °C, 60% relative humidity and a day and night cycles of 16–8 h. Another *O. surinamensis* population (strain OsNFRI) was obtained from the National Food Research Institute (Tsukuba, Japan) and used for genome sequencing.

**Elimination of *O. surinamensis* symbionts**. An *O. surinamensis* sub-population was treated for 12 weeks with tetracycline (150 mg/5g oat flakes) to eliminate their symbionts and then kept for several generations on a normal diet to exclude direct effects of tetracyclin on the host physiology[42]. Before the following experiments the aposymbiotic status of this beetle sub-population was confirmed. Therefore, 10 female adult beetles were individually separated in a single jar with oat flakes to lay eggs, as were symbiotic beetles in parallel populations. After 4 weeks, the adult generation was removed before their offspring finished metamorphosis and DNA of these females extracted and the symbiont titer was analyzed by quantitative PCR (see below;[42]).

**Symbiont genome sequencing, assembly, and annotation**. Total DNA was isolated from 20 pooled adult abdomina (without wings) of *O. surinamensis* JKI using the Epicentre MasterPure™ Complete DNA and RNA Purification Kit (Illumina Inc., Madison, WI, USA) including RNase digestion. Short-read library preparation and sequencing was performed at the Max-Planck-Genome-center Cologne, Germany (SRR12881563–SRR12881566) on a HiSeq2500 Sequencing System (Illumina Inc., Madison, WI, USA). Two further libraries were created from *O. surinamensis* strain OsNFRI. For the first library the DNA was extracted by QIAamp DNA Mini Kit (Qiagen, Germany) from 210 bacteriomes dissected from 60 adults. The library was prepared using the Nextera XT DNA Library Preparation Kit (Illumina Inc., Madison, WI, USA) and sequenced on a MiSeq (Illumina Inc., Madison, WI, USA) of AIST (Japan). For the second library the DNA was extracted by QIAamp DNA Micro Kit (Qiagen, Germany) from 24 bacteriomes dissected from six adults (each individual beetle contains four separate bacteriomes). The library was prepared using the Nextera DNA Library Preparation Kit (Illumina Inc., Madison, WI, USA) and sequenced on a NovaSeq 6000 (Illumina Inc., Madison, WI, USA) of Novagen (China). Adaptor and quality trimming was

performed with Trimmomatic[102]. In addition, we used two publicly available metagenome libraries of *O. surinamensis* (SRR5279855 and SRR6426882).

Long-read sequencing (SRR12881567–SRR12881568) was performed on a MinION Mk1B Sequencing System (Oxford Nanopore Technologies (ONT), Oxford, UK). Upon receipt of flowcells, and again immediately prior to sequencing, the number of pores on flowcells was measured using the MinKNOW software (v18.12.9 and 19.05.0, ONT, Oxford, UK). Flowcells were replaced into their packaging, sealed with parafilm and tape, and stored at 4 °C until use. Library preparation was performed with the Ligation Sequencing Kit (SQK-LSK109, ONT, Oxford, UK) and completed libraries were loaded on a flowcell (FLO-MIN106D, ONT, Oxford, UK) following the manufacturer's instructions.

Quality-controlled long reads were mapped using a custom-made kraken2 database containing the publicly available genomes of Bacteroidetes bacteria[103,104] to filter beetle-associated sequences using the supercomputer Mogon of the Johannes Gutenberg-University (Mainz, Germany). Hybrid assembly of MinION and Illumina reads was performed using SPAdes (v3.13.0) with the default settings[105]. This resulted in ~70,000 contigs that were then binned using BusyBee Web[106], screened for GC content and taxonomic identity to Bacteroidetes bacteria, and additionally checked manually for tRNAs and ribosomal proteins of Bacteroidetes bacteria. In total, 13 contigs were extracted, which were then automatically annotated with RAST[107] using the app *Annotate Microbial Assembly* (RAST_SDK v0.1.1) on KBase[108]. The annotated contigs were plotted using CIRCOS[109] (v0.69-6) for the visualization of gene locations, GC content and coverage. Additionally, the completeness of the obtained genome was assessed with the app *Assess Genome Quality with CheckM—v1.0.18* in KBase[45].

**Phylogenetic analyses**. A phylogenetic tree for placement of the intracellular symbiont of *O. surinamensis* within the Bacteroidetes was reconstructed using the KBase app *Insert Set of Genomes Into Species Tree* v2.1.10 (SpeciesTreeBuilder v0.0.12) based on the FastTree2 algorithm[110], including 49 highly conserved Clusters of Orthologous Groups (COG) genes[111].

A phylogenetic tree of the *aroA* gene (which codes for the EPSPS enzyme in the shikimate pathway) from the symbiont of *O. surinamensis* to predict its sensitivity to glyphosate was performed according to Motta et al. [25]. Manually selected *aroA* sequences from plants, gut bacteria as well as several intracellular insect symbionts were obtained from Uniprot (UniProt Consortium 2019), translated and aligned using MUSCLE[112] (v3.8.425) implemented in Geneious Prime 2019 (v2019.1.3, https://www.geneious.com). Phylogenetic reconstruction was performed with FastTree[110] (v2.1.12) and PhyML[113] (v2.2.4) implemented in Geneious Prime 2019 (v2019.1.3, https://www.geneious.com) using the Jones–Taylor–Thorton model with 20 rate categories and an optimized Gamma20 likelyhood (FastTree) and 1000 bootstrap replicates (PhyML). The obtained trees were visualized using FigTree (v1.4.4, http://tree.bio.ed.ac.uk/software/figtree/).

**Comparison with other Bacteroidetes bacteria**. Previously published Bacteroidetes genomes were re-annotated with RAST[107,114] in KBase[108] to compare the bacteria and to estimate the genome-wide nucleotide sequence divergence level. Therefore, we identified single-copy orthologs in each genome pair using OrthoMCL[115] (v2.0) in KBase. KEGG categories were then assessed via GhostKOALA[116] (v2.2) of each gene's amino acid sequence. Heatmaps were visualized using the 'ComplexHeatmap' package in Rstudio (V 1.1.463 with R V3.6.3). CIRCOS[109] (v0.69-6) was used to link orthologous genes.

Genomes of Bacteroidetes bacteria and other bacteria described as cuticle supplementing symbionts were compared in KBase[108] in more detail. Therefore, all genomes were re-annotated with RAST[107] and used to classify all annotated genes according to the SEED Subsystem[117] using the app *View Function Profile for Genomes* (v1.4.0, SEED Functional Group: Amino Acids and Derivatives). The resulting raw count of genes with annotation was visualized as a heatmap using the function 'heatmap.2' in the 'ggplot' package in Rstudio (V 1.1.463 with R V3.6.3).

**Glyphosate and aromatic amino acid supplementation**. Eight treatments were prepared to assess the supplementation of chorismate by the symbiont to the host (Tables 2 and 3). For each treatment, jars were filled with 5 g finely ground oat flakes and different combinations of aromatic amino acids (1% w/w of each L-tyrosine, L-tryptophan and L-phenylalanine; Sigma-Aldrich, Germany) and glyphosate (0.1 or 1% w/w; Sigma-Aldrich, Germany). The glyphosate concentration of 0.1% and 1% (or 0.059 and 0.0059 mmol/g) was chosen based on the experiment by Snyder and Rio[24] on tsetse flies. The 10 and 20 mM glyphosate added to the tsetse flies bloodmeal correspond to 0.1595 and 0.319% w/w glyphosate based on a blood density of 1.06 g/cm³. Helander et al report 250 mg per 48 L of soil to be equivalent to the maximum recommended amount of glyphosate for agronomical applications which translates to 0.0004% based on a fertile soil density of 1.3 g/cm³[90]. Food with the different supplements was weighed, mixed with distilled water, and dried overnight at 50 °C in an incubator. Afterwards, the dry material was ground and filled into the jars, before 50 adult apo- or symbiotic *O. surinamensis* with undefined sex were added to each jar to lay eggs. After 4 weeks of incubation at standard conditions mentioned above, the adult beetles were removed, and the jars checked daily for adult offspring. Freshly hatched beetles were isolated in a 48-well plate with the corresponding manipulated diet and developed for 7 days until cuticle biosynthesis is largely completed in

**Table 2 Experimental treatments for assessing impact of symbiont elimination, glyphosate exposure, and dietary aromatic amino acid supplementation on cuticle traits and symbiont titers in *O. surinamensis*.**

| | sym | apo | 1% aromatic amino acids | 0.1% glyphosate | 1% glyphosate |
|---|---|---|---|---|---|
| A | x | | | | |
| B | x | | | x | |
| C | x | | | | x |
| D | x | | x | | |
| E | x | | x | x | |
| F | x | | x | | x |
| G | | x | | | |
| H | | x | x | | |

symbiotic beetles[43]. Then the beetles were stored at −80 °C for DNA extraction or fixated in 4% paraformaldehyde in PBS for histological analysis[118].

**Quantitative PCR**. DNA of 8–10 adult beetle abdomina (without wings) per treatment group (symbiotic and aposymbiotic parent generations and diet supplementation treatments), respectively, was isolated using the Epicentre MasterPure™ Complete DNA and RNA Purification Kit following the manufacturer's instruction (Illumina Inc., USA) to verify presence of the endosymbiont and evaluate the impact of glyphosate and amino acid addition on symbiont titer. Bacterial 16S rRNA copies were quantified via quantitative PCR (qPCR) from single adult abdomina of *O. surinamensis*. DNA was dissolved in 30 μL low TE buffer (1:10 dilution of 1× TE buffer: 10 mM Tris–HCl + 1 mM EDTA). qPCRs were carried out in 25 μl reactions using EvaGreen (Solis BioDyne, Estonia), including 0.5 μM of each primer and 1 μl template DNA. All reagents were mixed, vortexed and centrifuged in 0.1-mL reaction tubes (Biozym, 711200). To amplify a symbiont specific 16S rRNA gene fragment, the primers OsurSym_fwd2 (5'-GGCAACTCTGAACTAGCTACGC-3') and mod. CFB563_rev (5'-GCACCCTT-TAAACCCAAT-3') were used[42]. qPCR was carried out in a Rotor-Gene Q thermal cycler (Qiagen, Germany).

Standard curves with defined copy numbers of the 16S rRNA gene were created by amplifying the fragment first, followed by purification and determination of the DNA concentration via NanoDrop1000 (Peqlab, Germany). After determination of the DNA concentration, a standard containing $10^{10}$ copies/μL was generated and 1:10 serial dilutions down to $10^1$ copies/μL were prepared. 1 μL of each standard was included in a qPCR reaction to standardize all measurements.

**Analysis of cuticle traits**. First, we determined melanization single-blinded via the inverse of digital red color values of 8–10 beetles from each treatment group to evaluate the impact of glyphosate, aromatic amino acids, and symbiont elimination on cuticle formation. Photographs were taken with a Zeiss StereoDiscovery V8 dissection stereoscope (Zeiss, Germany) under identical conditions using ZENCore software (Zeiss, Germany). Average red values were measured within a circular area covering the thorax with Natsumushi[120] and transformed into inverse red values[16,42].

Further, we measured cuticle thickness single-blinded of 8–10 adult beetles per treatment group, which were fixated in 4% paraformaldehyde in PBS. These beetles were embedded in epoxy resin (Epon_812 substitute; Sigma-Aldrich, Germany), and 1 μm cross-sections of the thorax next to the second pair of legs were cut on a microtome (Leica RM2245, Wetzlar, Germany) with a diamond blade and mounted on silanized glass slides with Histokitt (Roth, Germany). Images to measure cuticle diameter were taken with an AxioImager Z2 (Zeiss, Germany) at ×200 magnification and differential interference contrast. Mean cuticle diameter was measured at one randomly chosen dorsal, ventral, and lateral position, respectively, with the ZEN 2 Blue software distance tool (v2.0.0.0, Zeiss, Germany).

**Amino acid extraction**. For both aposymbiotic and symbiotic *O. surinamensis* beetles, eight individuals for each of five different developmental stages were analyzed: 5th instar larvae (of unknown age), 1-day-old pupae, 5-day-old pupae, 1-day-old adults, and 7-day-old adults. Aged individuals were obtained by separating 5th instar larvae individually into a 48-well plate coated with Fluon (AGC Chemicals, UK) to avoid beetles to escape, filled with three oat flakes for provision, and daily monitoring for pupation or emergence of adults. Once they reached the

desired age, individuals were frozen until further analysis. Adults had their elytrae removed to analyze them separately and both were dried overnight at 50 °C.

The following extraction, derivatization, and analysis procedure of amino acids (AAs) were adapted from Perez-Palacios et al. [121]. For the extraction of free AAs in larvae, pupae and elytra-free adults, three stainless steel beads and 500 μL 0.1 M HCl (Merck, Germany) were added to each insect sample and homogenized in a mixer mill MM200 (Retsch, Germany) at 30 Hz for 6 min. Subsequently, samples were centrifuged at 5000 rpm for 15 min at 4 °C using a CT15RE microcentrifuge (Himac, Japan) and the supernatant transferred to glass vials. Pellet were kept for extraction of bound AAs (see below), supernatants containing free AAs were mixed with 300 μL acetonitril (Carl-Roth, Germany) for deproteinization and 5 μL L-norleucine (2.5 μmol/mL; Sigma, Germany), which was used as an internal standard. Following centrifugation at 10,000 rpm for 3 min, samples were transferred to new vials and dried in a Savant DNA 110 SpeedVac Concentrator (Thermo Fisher Scientific, Germany). For the extraction of protein-bound Aas, the aforementioned pellets of the remaining body were resuspended in 500 μL 0.1 M HCl and transferred to separate glass vials. Similarly, dissected elytra were homogenized as described above and transferred to separate glass vials. After these samples were dried, 12 open vials (Kimble, USA) were placed in a larger hydrolysis vial (Kimble, USA) and a mixture of 187.5 μL HPLC grade water (Carl-Roth, Germany), 62.5 μL saturated phenol solution (84 g/L, Carl-Roth, Germany), and 250 μL 12 M HCl were added to the bottom of the hydrolysis vial for gas-phase derivatization. To ensure an oxygen-free atmosphere, the hydrolysis vial was three times evacuated and aerated with argon and afterwards tightly sealed. The hydrolysis was performed for 24 h at 110 °C. After equilibration to room temperature, 200 μL HCl were added to the single sample vials, the supernatant transferred to a new GC vial, and the samples dried again for 10 min in a Speedvac evaporator (Thermo Scientific, Germany).

**Derivatization of amino acids**. Derivatization is needed to analyze AAs via gas chromatography-mass spectrometry (GC–MS). N-tert-butyldimethylsilyl-N-methyltriflouroacetamid (MTBSTFA, Sigma-Aldrich, Germany) was utilized to silylate AAs [121]: 50 μL MTBSTFA and 50 μL acetonitrile were added to all dried extracts of bound and free AAs, followed by derivatization at 100 °C for 1 h. Also, a mixture of 17 AAs (each 25 μmol/L: L-alanine, L-arginine, L-aspartic acid, L-glutamic acid, glycine, L-histidine, L-isoleucine, L-leucine, L-lysine, L-methionine, L-phenylalanine, L-proline, L-serine, L-threonine, L-tyrosine, L-valine; and 12.5 μmol/L L-cysteine, Sigma-Aldrich, Germany) was prepared and derivatized, also including L-norleucine as an internal standard to identify single AA derivatives. 1 μL was used for subsequent analysis.

**Amino acid analysis with gas chromatography–mass spectrometry (GC–MS)**. GC–MS analysis was performed with a Varian 240MS ion trap mass spectrometer coupled to a Varian 450 gas chromatograph (Agilent, USA), using external ionization for all analyses and splitless injection with 280 °C injector temperature. Initial temperature of the GC oven was 100 °C for 2 min, followed by steady increase by 25 °C per minute up to 300 °C and final isothermal hold for 5 min. The carrier gas helium had a constant flow of 1 mL/min through a DB-5ms column (30 m × 2.25 mm ID, 0.25 μm film Agilent, USA). Electron impact spectra were recorded with the ion source at 160 °C and ion trap at 90 °C and analyzed using the Varian MS Workstation Version 6.9.3. AA derivatives were identified using the fragmentation patterns and retention times according to Pérez-Palacios et al. [121] and the external standards. Quantification was carried out with two to four fragment ions to increase signal to noise ratio (L-alanine $m/z$ = 158 & 232 & 260 & 428;

 **11**

**Table 3 Results of Dunn's Test assessing the impact of symbiont infection status, aromatic amino acid supplementation, and glyphosate exposure on cuticle traits (thickness and melanization) and symbiont titers in one-week-old *O. surinamensis* adults.**

| | Z | r | Lower 95% | Δ median | Upper 95% | p | Adjusted p |
|---|---|---|---|---|---|---|---|
| *Cuticle thickness* | | | | | | | |
| A–B | **1.98** | **0.45** | **0.49** | **1.64** | **3.13** | **4.75E−02** | 1.33E−01 |
| A–C | **3.48** | **0.84** | **1.95** | **3.41** | **5.45** | **5.08E−04** | **4.74E−03** |
| B–C | 1.64 | 0.39 | −0.08 | 1.70 | 3.68 | 1.01E−01 | 2.01E−01 |
| A–D | 1.50 | 0.36 | −0.80 | 1.30 | 3.98 | 1.35E−01 | 2.36E−01 |
| B–D | −0.39 | −0.09 | −2.92 | −0.45 | 2.33 | 6.99E−01 | 8.16E−01 |
| C–D | −1.92 | −0.48 | −5.11 | −2.10 | 0.36 | 5.43E−02 | 1.38E−01 |
| A–E | 0.77 | 0.17 | −0.65 | 0.92 | 2.42 | 4.41E−01 | 5.88E−01 |
| B–E | −1.29 | −0.28 | −2.59 | −0.83 | 0.77 | 1.97E−01 | 3.06E−01 |
| C–E | **−2.89** | **−0.66** | **−5.24** | **−2.56** | **−0.74** | **3.85E−03** | **2.69E−02** |
| D–E | −0.82 | −0.19 | −3.79 | −0.58 | 2.15 | 4.12E−01 | 5.77E−01 |
| A–F | 2.05 | 0.47 | 0.37 | 2.06 | 3.90 | 4.08E−02 | 1.27E−01 |
| B–F | 0.07 | 0.01 | −1.44 | 0.41 | 2.14 | 9.48E−01 | 9.48E−01 |
| C–F | −1.58 | −0.37 | −4.00 | −1.24 | 0.66 | 1.14E−01 | 2.13E−01 |
| D–F | 0.45 | 0.11 | −2.39 | 0.71 | 3.46 | 6.54E−01 | 7.96E−01 |
| E–F | 1.36 | 0.30 | −0.52 | 1.12 | 3.33 | 1.74E−01 | 2.87E−01 |
| A–G | **4.26** | **0.98** | **3.17** | **4.61** | **6.20** | **2.02E−05** | **5.64E−04** |
| B–G | **2.34** | **0.52** | **1.25** | **2.93** | **4.36** | **1.91E−02** | **7.63E−02** |
| C–G | 0.57 | 0.13 | −0.94 | 1.02 | 2.83 | 5.70E−01 | 7.25E−01 |
| D–G | **2.60** | **0.61** | **0.81** | **3.50** | **5.64** | **9.41E−03** | **4.39E−02** |
| E–G | **3.69** | **0.81** | **2.07** | **3.56** | **5.53** | **2.24E−04** | **3.13E−03** |
| F–G | 2.28 | 0.51 | 0.84 | 2.57 | 4.71 | 2.27E−02 | 7.95E−02 |
| A–H | 1.70 | 0.39 | −0.11 | 1.60 | 3.18 | 8.99E−02 | 1.94E−01 |
| B–H | −0.29 | −0.07 | −1.79 | −0.21 | 1.42 | 7.69E−01 | 8.28E−01 |
| C–H | −1.92 | -0.45 | −4.15 | −2.07 | −0.21 | 5.50E−02 | 1.28E−01 |
| D–H | 0.11 | 0.03 | −2.38 | 0.03 | 2.70 | 9.13E−01 | 9.46E−01 |
| E–H | 0.99 | 0.22 | −1.16 | 0.65 | 2.58 | 3.22E−01 | 4.74E−01 |
| F–H | −0.36 | −0.08 | −2.57 | −0.60 | 1.72 | 7.19E−01 | 8.06E−01 |
| G–H | **−2.64** | **−0.59** | **−4.85** | **−3.18** | **−1.51** | **8.35E−03** | **4.67E−02** |
| *Cuticle melanization* | | | | | | | |
| A–B | 0.15 | 0.03 | −0.60 | 0.06 | 0.71 | 8.83E−01 | 9.51E−01 |
| A–C | **3.32** | **0.74** | **0.60** | **1.34** | **2.03** | **8.91E−04** | **4.99E−03** |
| B–C | **3.18** | **0.71** | **0.60** | **1.29** | **2.02** | **1.49E−03** | **6.95E−03** |
| A–D | 0.1 | 0.02 | −0.66 | 0.03 | 0.69 | 9.22E−01 | 9.56E−01 |
| B–D | −0.05 | −0.01 | −0.70 | −0.02 | 0.69 | 9.63E−01 | 9.63E−01 |
| C–D | **−3.14** | **−0.72** | **−2.06** | **−1.33** | **−0.59** | **1.70E−03** | **6.82E−03** |
| A–E | 0.67 | 0.15 | −0.41 | 0.25 | 0.91 | 5.01E−01 | 6.38E−01 |
| B–E | 0.53 | 0.12 | −0.42 | 0.24 | 0.70 | 5.98E−01 | 6.98E−01 |
| C–E | **−2.65** | **−0.59** | **-1.83** | **−1.12** | **−0.46** | **8.04E−03** | **2.81E−02** |
| D–E | 0.56 | 0.13 | −0.45 | 0.23 | 0.88 | 5.77E−01 | 7.02E−01 |
| A–F | 2.35 | 0.53 | 0.16 | 0.90 | 1.66 | 1.88E−02 | 5.86E−02 |
| B–F | 2.2 | 0.49 | 0.14 | 0.86 | 1.63 | 2.76E−02 | 7.74E−02 |
| C–F | −0.97 | −0.22 | −1.19 | -0.45 | 0.27 | 3.29E−01 | 4.61E−01 |
| D–F | 2.19 | 0.50 | 0.11 | 0.82 | 1.64 | 2.86E−02 | 6.67E−02 |
| E–F | 1.68 | 0.38 | −0.11 | 0.64 | 1.49 | 9.37E−02 | 1.54E−01 |
| A–G | **4.16** | **0.93** | **1.02** | **1.76** | **2.38** | **3.17E−05** | **8.89E−04** |
| B–G | **4.01** | **0.90** | **0.97** | **1.69** | **2.40** | **5.96E−05** | **8.34E−04** |
| C–G | 0.84 | 0.19 | −0.38 | 0.42 | 1.10 | 4.02E−01 | 5.36E−01 |
| D–G | **3.95** | **0.91** | **1.01** | **1.69** | **2.39** | **7.73E−05** | **7.21E−04** |
| E–G | **3.49** | **0.78** | **1.04** | **1.40** | **2.18** | **4.86E−04** | **3.40E−03** |
| F–G | 1.81 | 0.40 | −0.05 | 0.76 | 1.56 | 6.99E−02 | 1.22E−01 |
| A–H | 2.19 | 0.49 | 0.08 | 0.85 | 1.65 | 2.83E−02 | 7.21E−02 |
| B–H | 2.05 | 0.46 | 0.04 | 0.84 | 1.53 | 4.07E−02 | 8.77E−02 |
| C–H | −1.13 | -0.25 | −1.22 | −0.48 | 0.32 | 2.58E−01 | 3.80E−01 |
| D–H | 2.04 | 0.47 | 0.01 | 0.79 | 1.60 | 4.16E−02 | 8.33E−02 |
| E–H | 1.52 | 0.34 | −0.17 | 0.68 | 1.43 | 1.28E−01 | 1.99E−01 |
| F–H | −0.16 | −0.04 | −0.83 | 0.01 | 0.85 | 8.76E−01 | 9.81E−01 |
| G–H | −1.97 | −0.44 | −1.62 | −0.79 | −0.18 | 4.90E−02 | 9.15E−02 |
| *Symbiont 16S titer* | | | | | | | |
| A–B | 1.01 | 0.23 | 20,211 | 149,112.5 | 286891 | 3.10E−01 | 3.62E−01 |
| A–C | **3.93** | **0.88** | **275,297** | **380,359** | **546,505** | **8.61E−05** | **3.44E−04** |
| B–C | **2.91** | **0.65** | **101162** | **259,656.5** | **360,640** | **3.57E−03** | **9.10E−03** |
| A–D | 1.51 | 0.35 | 93,032 | 218,182.5 | 356,475 | 1.31E−01 | 1.94E−01 |
| B–D | 0.52 | 0.12 | −56,233 | 75,219 | 198341 | 6.02E−01 | 6.49E−01 |

**Table 3 (continued)**

|     | Z | r | Lower 95% | Δ median | Upper 95% | p | Adjusted p |
|-----|------|-------|-----------|-----------|-----------|----------|------------|
| C–D | **−2.32** | **-0.53** | **−274,985** | -186004.5 | **−85,291** | **2.06E−02** | **3.84E−02** |
| A–E | **2.48** | **0.55** | **183,686** | **308,129** | **417,996** | **1.29E−02** | **2.79E−02** |
| B–E | 1.47 | 0.33 | 59,841 | 134,995.5 | 26,9973 | 1.41E−01 | 1.97E−01 |
| C–E | −1.44 | -0.32 | −176,605 | −114,384 | −16,320 | 1.49E−01 | 1.99E−01 |
| D–E | 0.91 | 0.21 | 658 | 70,294.5 | 171,885 | 3.62E−01 | 4.05E−01 |
| A–F | **4.07** | **0.91** | **279,605** | **401,114** | **545,438** | **4.64E−05** | **2.60E−04** |
| B–F | **3.06** | **0.68** | **137,507** | **259,587.5** | **366,228** | **2.21E−03** | **6.20E−03** |
| C–F | 0.15 | 0.03 | −66,054 | −46 | 68,617 | 8.80E−01 | 8.83E−01 |
| D–F | **2.46** | **0.56** | **88,604** | **185,935.5** | **273,826** | **1.39E−02** | **2.79E−02** |
| E–F | 1.59 | 0.36 | 26,819 | 114,969 | 176,630 | 1.12E−01 | 1.84E−01 |
| A–G | **5.36** | **1.20** | **310,395** | **456,309** | **550,371** | **8.14E−08** | **1.14E−06** |
| B–G | **4.35** | **0.97** | **244,565** | **288,187** | **432,302** | **1.35E−05** | **9.50E−05** |
| C–G | 1.44 | 0.32 | −9 | 10,947 | 80,717 | 1.50E−01 | 1.91E−01 |
| D–G | **3.71** | **0.85** | **182,731** | **216,825.5** | **306,114** | **2.04E−04** | **7.14E−04** |
| E–G | **2.88** | **0.64** | **114,693** | **163089.5** | **183,514** | **3.98E−03** | **9.30E−03** |
| F–G | 1.29 | 0.29 | 3 | 11,905 | 89,556 | 1.96E−01 | 2.39E−01 |
| A–H | **5.61** | **1.25** | **318,871** | **45,5935.5** | **557,535** | **2.05E−08** | **5.74E−07** |
| B–H | **4.59** | **1.03** | **253,041** | **294,578** | **432,331** | **4.34E−06** | **4.05E−05** |
| C–H | 1.68 | 0.38 | 17 | 10,980 | 80,746 | 9.28E−02 | 1.62E−01 |
| D–H | **3.95** | **0.91** | **181,983** | **216,855.5** | **306,115** | **7.78E−05** | **3.63E−04** |
| E–H | **3.12** | **0.70** | **115,217** | **162,716** | **189,619** | **1.79E−03** | **5.57E−03** |
| F–H | 1.53 | 0.34 | 5 | 11938 | 89585 | 1.24E−01 | 1.94E−01 |
| G–H | 0.24 | 0.05 | −775 | 18.03559 | 8514 | 8.07E−01 | 8.37E−01 |

Significant results (adjusted $p < 0.05$) are highlighted in bold. Z-test statistic, effect size r, lower and upper boundary of 95% confidence intervals and difference of medians, unmodified and Benjamini–Hochberg corrected p-values are reported. Significant results (P.adj < 0.05) are highlighted in bold. A: Sym; B: Sym + 0.1% G; C: Sym + 1% G; D: Sym + 1% AA; E: Sym + 1% AA + 0.1% G; F: Sym + 1% AA + 1% G; G: Apo; H: Apo + 1% AA.

L-arginine $m/z = 199$ & 442; L-aspartic acid $m/z = 302$ & 316 & 390 & 418; L-cystine $m/z = 58$ & 341 & 442; L-glutamic acid $m/z = 272$ & 286 & 359 & 432; glycine $m/z = 189$ & 218 & 246; L-histidine $m/z = 196$ & 280.5 & 338.5 & 440.5, L-isoleucine $m/z = 200$ & 274 & 302; L-leucine $m/z = 200$ & 274 & 302; L-lysine $m/z = 198$ & 272 & 300; L-methionine $m/z = 218$ & 292 & 320; L-nor-leucine $m/z = 200$ & 274 & 302; L-phenylalanine $m/z = 234$ & 392 & 308 & 336; L-proline $m/z = 184$ & 258 & 286; L-serine $m/z = 288$ & 362 & 390; L-threonine $m/z = 303$ & 376 & 404; L-tyrosine $m/z = 302$ & 364 & 438 & 466; L-valine $m/z = 186$ & 260 & 288).

**Statistical analyses and reproducibility**

*Reproducibility.* Sequencing experiments are based on multiple extracts/libraries, generated from the same stock culture, but also a different culture as well as a publicly deposited sequence library. Multiple independent individuals or their offspring were generated in each experiment and subjected to statistical analyses, we aimed at sample sizes of 8–10 individuals, however in some cases some were lost during processing. Replicates were defined by individuals raised in different containers kept under identical conditions but seeded with different parents from the same stock population.

*Analysis of symbiont titer and cuticular traits.* Influence of amino acids and glyphosate on the symbiont titer, cuticle thickness and melanization of the adult beetles was analyzed using Dunn's test from the package 'FSA' in RStudio (V 1.1.463 with R V3.6.3) with two-sided, for multiple testing corrected post-hoc tests using the Benjamini–Hochberg method[119] implemented in the R 'stats' package. Effect size r was calculated manually from R output following the formula $r = Z/\sqrt{N}$. Plots were visualized using 'ggplot2'.

*Analysis of amino acid titers.* The single amino acid measurements were first normalized by the internal standard to account for deviations in amount or concentration during the analysis procedure and further normalized by the total amount of all amino acids to account for differences in body size between life stages but also animals with and without symbionts. The single normalized amino acid measurements were transformed to an approximate Gaussian distribution. The transformation was chosen using the powerTransform command from the package 'car' (alanin: negative square root, arginine: cubic root, aspartic acid: decadic logarithm, glutamic acid: square root, glycine: cubic root, isoleucine: decadic logarithm, leucine: decadic logarithm, lysine: square root, ornithine: square root, phenylalanine: square root, proline: decadic logarithm, serine: cubic root, threonine: decadic logarithm, tyrosine: square root, valine: cubic root; the amino acids cystine, histidine and methionine were not detected).

Symbiont influence on the titer of the single amino acids was analyzed with separate generalized linear models with life stage (larva, two pupal stages, two adult stages), amino acid source (free amino acids in body, protein-bound amino acids in body, total amino acids in elytrae) and symbiont presence/absence as factors, allowing for interactions between all of them using the command glm from the 'stats' package in R Studio (V 1.1.463 with R V3.6.3). P-values were corrected for multiple testing following the classical Bonferroni[122]. In case of significant symbiont influence, we conducted pairwise post-hoc tests between symbiotic and aposymbiotic samples for each life stage/amino acid source using unpaired Wilcoxon rank-sum tests including Bonferroni correction[122] from the R 'stats' package to identify the specific life stage/amino acid source of symbiont influence. Effect size r was calculated manually from R output following the formula $r = Z/\sqrt{N}$. Plots were visualized using 'ggplot2'.

**Reporting summary.** Further information on research design is available in the Nature Research Reporting Summary linked to this article.

## Data availability

Sequencing libraries and the assembles genome of the *Oryzaephilus surinamensis* symbiont (proposed *Candidatus* Shimatogenerans silvanidophilus OSUR) were uploaded to the DNA Databank of Japan (accession numbers DRA010986 and DRA010987), the NCBI Sequence Read Archive (accession numbers SRR12881563–SRR12881566) and Genbank (JADFUB000000000). Raw data of quantitative cuticle measurements are available at the data repository of the Max Planck Society 'Edmond'[123].

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

## Acknowledgements

The authors thank Dagmar Klebsch and Rebekka Janke for valuable technical assistance, Cornel Adler for the original provisioning of an O. surinamensis culture, John McCutcheon and Piotr Łukasik for their initial input on our genome sequencing approach, the Johannes Gutenberg University Mainz for computation time granted on the supercomputer Mogon, Christian Meesters' administrative assistance on Mogon and Minoru Moriyama's support for genome sequencing. The authors further acknowledge financial support of the Johannes Gutenberg University Mainz (intra-mural funding to T.E.), a Consolidator Grant of the European Research Council (ERC CoG 819585 "SYMBeetle" to M.K.), the Japan Science and Technology Agency ERATO Grant (JPMJER1803 and JPMJER1902 to T.F.), the Japan Society for the Promotion of Science KAKENHI Grant (20J13769 to B.H. and 17H06388 to T.F.), and the Max-Planck-Society (to T.E., B.W., and M.K.). T.E. also acknowledges the stimulating

International Symbiosis Society meeting 2018 in Oregon, especially inspiring presentations by Rita Rio and Joel Sachs.

## Author contributions
T.E. and M.K. conceived the study. J.S.T.K., E.B., B.H. and T.E. sequenced and assembled the symbiont genome. J.S.T.K. and E.B. annotated the genomes and performed symbiont genomic analysis and J.S.T.K. and T.E. performed phylogenetic analyses of the symbionts. S.B., J.C.W. and T.E. performed amino acid analysis. J.K. and B.W. performed dietary supplementation experiments and analysis. J.S.T.K. and T.E. wrote the paper, with input from M.K. and T.F. All authors read and commented on the manuscript.

## Funding

## Competing interests
The authors declare no competing interests.
