## [Peer Review File · Communications Biology]

Reviewers' comments:

Reviewer #1 (Remarks to the Author):

This is an interesting and well-structured study on the potential effect of glyphosate on an important endosymbiont with cascading negative effects on cuticle sclerotization. There has been much attention on the hitherto unrecognized role of glyphosate on the ability of insects to complete melanization and sclerotization of the cuticle. This study contributes to this interesting area of research by providing information on an endosymbiotic bacterium and how it is negatively affected by glyphosate mainly via inhibition of the shikimate pathway. The present manuscript is following up on a previous study that outlined the fitness consequences of the thinner cuticle observed in hosts lacking the endosymbiont.

The authors provide a neat study model that allows them to remove the endosymbiont without compromising host survival. Thus, they experimentally verify that cuticle thickness is reduced in individuals lacking the endosymbiont, and they indicate the role of tyrosine by partially restoring cuticle thickness when adding a dietary supplement containing tyrosine.

I find that the manuscript is well written and presented. My comments/questions are mainly related to clarification of methods and the interpretation of the results.

1. You emphasize on the role of tyrosine in insect cuticle sclerotization. However, tyrosine is also instrumental in insect immunity via the phenoloxidase pathway. Wouldn't you think that glyphosate potentially could have a strong negative effect on insect fitness by reducing melanization of potential pathogens? That obviously could be tested simply by experimentally exposing the insect to an entomopathogen. In the case of pathogenic fungi penetrating the cuticle it would have a double effect by increasing cuticle penetrability by the fungus and reducing the phenoloxidase activity of the insect. I think that would be worth pursuing, and the potential negative effect on melanization by glyphosate probably should be addressed.

2. As you mentioned in the Introduction, glyphosate is capable of modulating the gut microbiome in honeybees. Given the importance of the gut microbiome in health and normal development, changes to the composition of the microbiome may provide an additional (or alternative) explanation to the observed effects, and I think that deserves to be mentioned in the Discussion.

3. You measured cuticle thickness on day 7. Is it possible that sclerotization/melanization was delayed rather than inhibited? In other words, do you have data to verify that cuticle thickness in glyphosate treated individuals didn't catch up with control individuals on day 8 or 9?

4. You used 0.1 and 1% glyphosate. How do those concentrations link to field-relevant concentrations that insects would experience in real life? The alarming scenario of glyphosate being related to recent insect declines depends on whether insects would experience concentrations that are high enough to severely affect endosymbiont presence and abundance.

Minor corrections/typos

Line 49: ...animals benefit from the more specialized... ("the" instead of "a")

Line 98-102: As mentioned above a thinner cuticle could also increase mortality related to fungal attack.

Line 164: space between pathway and except

Results (general): Be careful not to interpret results in the results section. That belongs in the Discussion

Line 268: lacks "d" in feedback

Line 271: Give p-value as $P < 0.0001$ rather than $p = 0$

Line 280: typo: phylogenetic rather than phylogentetic

Line 376: insert "that" instead of "the" before "...either the symbionts can not grow..."

Line 509: I find it a little confusing that you use the word "infection" here if you are talking about a non-pathogenic endosymbiont. Maybe rephrase to call it verify presence of the endosymbiont. Table 1: I like the visualization of the experimental design. However, it looks like you made a mistake in treatment E where you should have had an x in 1% aromatic amino acids and no x in 1% glyphosate (please check).

Reviewer #2 (Remarks to the Author):

Kiefer et al. characterized the genome of a bacterial nutritional endosymbiont as it pertains to proper cuticular development of the sawtoothed grain beetle *Oryzaephilus surinamensis*. Since the symbiont genome was concluded to produce precursors of cuticle synthesis for *O. surinamensis*, feeding bioassays were conducted to assess the effect of incorporating aromatic amino acids and the herbicide glyphosate on cuticle development of symbiotic and asymbiotic beetles. The dietary bioassays revealed that cuticle development in asymbiotic beetles could be partially rescued by supplementing with tyrosine. Exposure of *O. surinamensis* during larval development to glyphosate also impaired the establishment of the symbiont, and disrupted cuticle development. The data analysis and interpretation of results appear to be reasonable. However, an important limitation of the existing manuscript is the concentration of glyphosate tested in the study (1% and 0.1% w/w) should be described in less ambiguous units, for instance, mg/L or ml/L. For instance, Snyder and Rio (2015) incorporated glyphosate into tsetse fly diet at 10 and 20 Mm, while Motta and Moran (2020) tested glyphosate at 0.01 mM to 1.0 mM. Motta et al. (2018) tested glyphosate at 5 mg/L and 10 mg/L on honey bees to mimic environmental levels of 1.4 to 7.6 mg/L. It should be addressed in the manuscript if environmentally relevant concentrations were tested.

Line 252-254: Clarify what analyses the p values are reporting. Thickness and melanization of the cuticle?

Line 256: Were asymbiotic beetles exposed to glyphosate?

Fig 3: Are boxplots included in Fig 3 as in Fig 5? The resolution of Fig 3 is not very good and difficult to interpret. A description of the graph type presented in Fig 3 should be provided in the caption, for instance, the basis for the outlines around data for each treatment.

Fig 5: Details about the boxplots should be provided, for instance, defining the circles and box parameters for each treatment. As with Fig 3, the basis for the outlines should be described, including the upper and lower vertical points of the outlines.

Reviewer #3 (Remarks to the Author):

The manuscript titled " Inhibition of a nutritional endosymbiont by glyphosate abolishes 2 mutualistic benefit on cuticle synthesis " deals with the description on the contribution of the beetle *Oryzaephilus surinamensis*, the proposed 'Candidatus *Shikimatogenerans silvanidophilus* 196 OSUR', to the beetle titers in the aromatic amino acid tyrosine and its importance for cuticle sclerotization and melanization, and how glyphosate impacts this endosymbiont and the cuticle of the beetle, which support the already claimed idea of the important indirect impact of the use of glyphosate on insects. Moreover, the paper includes the obtaining, annotation and analysis of the genome sequence of the bacterial symbiont; based on this, it describes the metabolism of the symbiont and some potential roles within its host, such as the provision of precursors for cuticle synthesis via the shikimate pathway, and yet includes a phylogenetic analysis of EPSPS enzymes from different insect endosymbionts, including both, glyphosate sensitive and tolerant enzymes.

Thus, the manuscript presents nice and interesting novel information. These considerations make the paper interesting for the readership of the Communications Biology journal.

But the abstract, the Ms is well written and structured, although it could be synthesized in some parts. Some details are missing in the materials and methods section.

Specific comments are the following:

Along the whole document: Latin names (taxa) should be written in italics, as well as gene names (some examples are given below)

Abstract: from my point of view, this is the poorest part of the article and it could be rewritten for clarity. For instance:

Lines 25-27 "decreased titers of the aromatic aminoacid tyrosine in symbiont-depleted beetles support the ability to synthesize phenate..." ????. Maybe you mean "decreased titers of the aromatic aminoacid tyrosine in symbiont-depleted beetles support the ability of symbionts to synthesize phenate..."

Lines 28-29: "Glyphosate exposure inhibited symbiont establishment during development", which development??

Introduction:

Line 89: *Bacteroidetes* in italics

Line 90: *Coleoptera*, *Silvanidae* in italics

Results:

Lines 116-119: it should be explained better why you focus on the *Bacteroidetes* endosymbiont. Do you get sequences only from one bacterium within the bacteriome??

Lines 124 and 125: when you write kbp, it should be bp

Line 143: endosymbiont should be endosymbionts

Line 158: this section title is a bit odd. Written like this it seems that *O. surinamensis* has just one symbiont (I am not sure if that is the case). In any case, why not to use already the proposed symbiont name? Something like: *Candidatus Shikimatogenerans silvanidophilus* 196 OSUR encodes glycolysis and shikimate pathways. Just a suggestion...

Lines 168 and 169: *Carsonella rudii* and *Nardonella* in italics

Line 195: *Shikimatogenerans silvanidophilus* in italics

Line 196: for "the symbiont" should be "for this symbiont".

Lines 200-202: I do not understand what you mean with this.

Lines 209-215: too much explanation for a results section.

Line 243: the gene name (*aro*) should be in italics.

Lines 179-184: I miss a short explanation to the reason to do this.

Discussion:

Lines 302: gene names should be in italics

Materials and methods:

"Elimination of *O. surinamensis* symbionts" section: details such as the antibiotic concentration are missing.

"Symbiont genome sequence, assembly and annotation": it is not explained at all how you get X "bacteriomes dissected". I do not understand at all what a bacteriome dissection is, and, in any case, it is not explained how it was obtained.

Line 463: name of the gene in italics

"Comparison of bacteria": this section title is too broad. Please, choose a title representative of what you explain in this section.

Supplementary Table 5 contains the most important information of the article, according to the way the work has been presented, since it shows how glyphosate impacts the beetle's

development. Thus, I would say that it should not be a "supplementary file" and, in contrast, be included with the main text.

Dear Reviewers,

Thank you very much for your valuable feedback. Please find our response to all raised questions and comments below, highlighted in red. Changed figures and added text in the main manuscript are included after the corresponding question/comment below.

Reviewer #1 (Remarks to the Author):

This is an interesting and well-structured study on the potential effect of glyphosate on an important endosymbiont with cascading negative effects on cuticle sclerotization. There has been much attention on the hitherto unrecognized role of glyphosate on the ability of insects to complete melanization and sclerotization of the cuticle. This study contributes to this interesting area of research by providing information on an endosymbiotic bacterium and how it is negatively affected by glyphosate mainly via inhibition of the shikimate pathway. The present manuscript is following up on a previous study that outlined the fitness consequences of the thinner cuticle observed in hosts lacking the endosymbiont.

The authors provide a neat study model that allows them to remove the endosymbiont without compromising host survival. Thus, they experimentally verify that cuticle thickness is reduced in individuals lacking the endosymbiont, and they indicate the role of tyrosine by partially restoring cuticle thickness when adding a dietary supplement containing tyrosine.

I find that the manuscript is well written and presented. My comments/questions are mainly related to clarification of methods and the interpretation of the results.

1. You emphasize on the role of tyrosine in insect cuticle sclerotization. However, tyrosine is also instrumental in insect immunity via the phenoloxidase pathway. Wouldn't you think that glyphosate potentially could have a strong negative effect on insect fitness by reducing melanization of potential pathogens? That obviously could be tested simply by experimentally exposing the insect to an entomopathogen. In the case of pathogenic fungi penetrating the cuticle it would have a double effect by increasing cuticle penetrability by the fungus and reducing the phenoloxidase activity of the insect. I think that would be worth pursuing, and the potential negative effect on melanization by glyphosate probably should be addressed.

Thank you, this is an excellent question. We are currently investigating the impact of this symbiosis on pathogen and predator defense. While tyrosine availability probably also impacts phenoloxidase activity, tyrosine seems to be mostly limited during the metamorphosis and in the few days during cuticle maturation of adults. Later in life tyrosine seems to be not or less limited, also supported by the quantification of tyrosine titers in aposymbiotic vs symbiotic adults at day 7 in this manuscript. Thus, we anticipate that even the defensive aspect is largely driven by the cuticle itself. We are actually about to submit a manuscript on the impact on defense against natural enemies conferred by this symbiosis.

2. As you mentioned in the Introduction, glyphosate is capable of modulating the gut microbiome in

honeybees. Given the importance of the gut microbiome in health and normal development, changes to the composition of the microbiome may provide an additional (or alternative) explanation to the observed effects, and I think that deserves to be mentioned in the Discussion.

It is indeed interesting how much a less specialized gut microbiota might contribute to functions that are also supported by highly specialized, bacteriome associated symbionts, especially as this question has not drawn much attention in the field. However, based on the high degree of specialization of the genome of *Candidatus Shikimatogenerans silvanidophilus* and its high abundance in the bacteriomes we would predict that it contributes the vast majority of microbially supplemented tyrosine, especially as no extracellular, vertically transmitted microbiota was described so far. E.g. Fluorescence in situ hybridization of eggs and also adults in Engl et al. 2018 and this study did not reveal a recognizable amount of egg surface transmitted or gut retained microbes. The gut microbiota is thus probably acquired environmentally and should not differ between aposymbiotic and symbiotic control groups. However, we are currently investigating this point in depth with the additional focus of potential feedback of the symbiont presence on the development of structural traits of the gut itself and the establishment of the gut microbiota. Thus, we would refer the reviewer to this upcoming study and not add another subsection, that will in our opinion distract from the logic behind the current discussion.

3. You measured cuticle thickness on day 7. Is it possible that sclerotization/melanization was delayed rather than inhibited? In other words, do you have data to verify that cuticle thickness in glyphosate treated individuals didn't catch up with control individuals on day 8 or 9?

We do not have data in this manuscript on later life stages. However, the melanization of the cuticle is an ongoing process in *O. surinamensis*, while its general synthesis in terms of thickness is finished during early adult development. Thus, melanization could increase with age, but not cuticle thickness. In our previous study (Engl et al. 2018) we analyzed cuticle traits of randomly selected individual symbiotic and aposymbiotic beetles of an age between 0-12 weeks and still found systematic differences between both treatments, suggesting there is not only a delayed, but maintained cuticle deficiency. However, the susceptibility towards stresses, including desiccation, but also natural enemies is probably highest in the first days of adults until cuticle synthesis is largely concluded. Thus, we will include a fine scale analysis in the upcoming study on susceptibility to natural enemies.

4. You used 0.1 and 1% glyphosate. How do those concentrations link to field-relevant concentrations that insects would experience in real life? The alarming scenario of glyphosate being related to recent insect declines depends on whether insects would experience concentrations that are high enough to severely affect endosymbiont presence and abundance.

Please see response to reviewer 2's similar, but more extensive question.

Minor corrections/typos

Line 49: ...animals benefit from the more specialized... ("the" instead of "a") **done**

Line 98-102: As mentioned above a thinner cuticle could also increase mortality related to fungal attack.

Line 164: space between pathway and except **done**

Results (general): Be careful not to interpret results in the results section. That belongs in the Discussion

Line 268: lacks "d" in feedback **done**

Line 271: Give p-value as $P < 0.0001$ rather than $p = 0$ **done**

Line 280: typo: phylogenetic rather than phylogentetic **done**

Line 376: insert "that" instead of "the" before "...either the symbionts can not grow..." **done**

Line 509: I find it a little confusing that you use the word "infection" here if you are talking about a non-pathogenic endosymbiont. Maybe rephrase to call it verify presence of the endosymbiont. **done**

Table 1: I like the visualization of the experimental design. However, it looks like you made a mistake in treatment E where you should have had an x in 1% aromatic amino acids and no x in 1% glyphosate (please check). **done**

Reviewer #2 (Remarks to the Author):

Kiefer et al. characterized the genome of a bacterial nutritional endosymbiont as it pertains to proper cuticular development of the sawtoothed grain beetle *Oryzaephilus surinamensis*. Since the symbiont genome was concluded to produce precursors of cuticle synthesis for *O. surinamensis*, feeding bioassays were conducted to assess the effect of incorporating aromatic amino acids and the herbicide glyphosate on cuticle development of symbiotic and asymbiotic beetles. The dietary bioassays revealed that cuticle development in asymbiotic beetles could be partially rescued by supplementing with tyrosine. Exposure of *O. surinamensis* during larval development to glyphosate also impaired the establishment of the symbiont, and disrupted cuticle development. The data analysis and interpretation of results appear to be reasonable. However, an important limitation of the existing manuscript is the concentration of glyphosate tested in the study (1% and 0.1% w/w) should be described in less

ambiguous units, for instance, mg/L or ml/L. For instance, Snyder and Rio (2015) incorporated glyphosate into tsetse fly diet at 10 and 20 Mm, while Motta and Moran (2020) tested glyphosate at 0.01 mM to 1.0 mM. Motta et al. (2018) tested glyphosate at 5 mg/L and 10 mg/L on honey bees to mimic environmental levels of 1.4 to 7.6 mg/L. It should be addressed in the manuscript if environmentally relevant concentrations were tested.

As *O. surinamensis* does not feed on a liquid substrate like Tsetse flies or honeybees, using concentration values referencing a liquid base is in our opinion somewhat awkward, while stating solid in solid concentrations using defined "w/w%" (L495-508) will create less conversion problems to a trained natural scientist (see below). To accommodate this difference in substrates we added a comment in both the method and discussion section, clarifying the used concentrations and relating to above mentioned studies

(L373: "Concordantly, our results on an intracellular beetle symbiosis show that exposure to agronomically applied⁹⁰ or previously tested glyphosate levels^{24,25} decreases symbiont titers and recapitulates cuticular phenotypes of aposymbiotic beetles, indicating that the inhibition of the symbiont's shikimate pathway results in aromatic amino acid starvation of both host and symbiont."

& L501-507: "The glyphosate concentration of 0.1 and 1 % (or 0,059 and 0,0059 mmol/g) was chosen based on the experiment by Snyder and Rio²⁴ on tsetse flies. The 10 and 20 mM glyphosate added to the tsetse flies bloodmeal correspond to 0.1595 and 0.319 w/w% glyphosate based on a blood density of 1.06 g/cm³. Similarly, Helander et al report 250g per 48L of soil to be equivalent to the maximum recommended amount of glyphosate for agronomical applications which translates to 0.4 % based on a fertile soil density of 1.3 g/cm³⁹⁰.").

To briefly summarize our chosen concentration, we originally based our calculation on the study Snyder and Rio (2015). The studies by Motta et al and Motta and Moran were not published at that time. 10mM (=10mmol/L) of glyphosate added to the blood meal of tsetse flies corresponds to 1691mg/L or 1.691g/L based on the molecular weight of 169,1 g/mol of glyphosate. Based on average blood density (as this value was not reported by Snyder and Rio) of 1.06g/cm³ (=kg/L), this corresponds to 1.595g/kg or 0.1595%. The 20mM treatment corresponds accordingly to 0.319%.

Thus, both of our treatments of 0.1 and 1% glyphosate in oat flower represent a similar concentration.

Based on the study by Helander et al (2019: Glyphosate residues in soil affect crop plant germination and growth) 250mg of glyphosate per 48L soil represents the recommended maximal agronomical applied amount of glyphosate. Again, using a volume to measure the substrate is problematic, as the density of soil was not reported. Assuming a density of "healthy" soil of $\sim 1.3 \text{ g/cm}^3$, this corresponds to 4g/kg or 0.4%, also within our chosen concentration range.

It is worth mentioning, that this represents not residual amounts of glyphosate in the environment, but the actually applied amounts in agricultural practices. Effects of long-term exposure to lower amounts of glyphosate that correspond residues in the environment on intracellular symbionts, especially transgenerational effects, are currently under investigation within our group.

Line 252-254: Clarify what analyses the p values are reporting. Thickness and melanization of the cuticle?

Done – we added "thickness" and "melanization" in brackets behind the p-values

Line 256: Were asymbiotic beetles exposed to glyphosate?

No, we did not expose aposymbiotic beetles to glyphosate.

Fig 3: Are boxplots included in Fig 3 as in Fig 5? The resolution of Fig 3 is not very good and difficult to interpret. A description of the graph type presented in Fig 3 should be provided in the caption, for instance, the basis for the outlines around data for each treatment.

The original figures 3, 5 and Supplementary Figure 3 contained boxplots within the violin plots. We agree that the resolution was in some graphs problematic. We thus simplified the graphs by removing the box plots, depicting only the individual datapoints, the data distribution by a violin outline and the median with a horizontal line. We added the following, additional comment in the figure captions explaining the visualization: "The data distribution is visualized with violin plots and an additional horizontal line depicting the median."

The novel figures including captions are as follow. On this note, we found some transposed digits in the measurement of the cuticle thickness of the "symbiotic beetles supplemented with 0.1% glyphosate" group. The statistical results in L250-257, Table 2 and Figure 5 contain the updated results. The original measurements are contained in the deposited raw data of quantitative measurements under <https://dx.doi.org/10.17617/3.5l>.

Figure 3. Comparison of titers of the three amino acid tyrosine, proline and glutamic acid that that were influenced by symbiont presence. Shown are free amino acid titers in the whole body (without elytrae in case of adults) of symbiotic and aposymbiotic *O. surinamensis* beetles. Red: aposymbiotic beetles, Blue: symbiotic beetles. The data distribution is visualized with violin plots and an additional horizontal line depicting the median. The FDR corrected unpaired, two-sided Wilcoxon-rank-sum-tests: ns $p > 0.05$, * $0.05 < p < 0.01$, ** $p < 0.01$.

Figure 5. Effect of glyphosate exposure and aromatic amino acid supplementation on cuticle traits and symbiont titers in symbiotic and aposymbiotic beetles. Cuticle thickness (a), melanization measured as thorax coloration (b), and symbiont titers (c) of aposymbiotic and symbiotic adults reared on different food compositions. The data distribution is visualized with violin plots and an additional horizontal line depicting the median. Different letters indicate significant differences between experimental treatments (Dunn's Test, $\alpha \leq 0.05$).

Supplementary Figure 3. Comparison of titers of the three amino acid tyrosine, proline and glutamic acid that were influenced by symbiont presence. Shown are (a) bound amino acid titers in the whole body (without elytrae in case of adults) and (b) combined free and bound titers in adult elytrae of symbiotic and aposymbiotic *O. surinamensis* beetles. Red: aposymbiotic beetles, Blue: symbiotic beetles. The data distribution is visualized with violin plots and an additional horizontal line depicting the median. FDR corrected unpaired, two-sided Wilcoxon-rank-sum-tests: ns $p > 0.05$, * $0.05 < p < 0.01$, ** $p < 0.01$.

Fig 5: Details about the boxplots should be provided, for instance, defining the circles and box parameters for each treatment. As with Fig 3, the basis for the outlines should be described, including the upper and lower vertical points of the outlines.

We do report results of these statistical tests within the text of the experimental section as well as in a categorized manner in Figure 5. Due to the extent of the detailed result table, we thought it better to provide it as a separate file, similar as supplementary tables. However, in response to reviewer 3 we now included them as tables 2 and 3 in the main manuscript.

Reviewer #3 (Remarks to the Author):

The manuscript titled "Inhibition of a nutritional endosymbiont by glyphosate abolishes 2 mutualistic benefit on cuticle synthesis" deals with the description on the contribution of the beetle *Oryzaephilus surinamensis*, the proposed 'Candidatus Shikimatogenerans silvanidophilus 196 OSUR', to the beetle titers in the aromatic amino acid tyrosine and its importance for cuticle sclerotization and melanization, and how glyphosate impacts this endosymbiont and the cuticle of the beetle, which support the already claimed idea of the important indirect impact of the use of glyphosate on insects. Moreover, the paper includes the obtaining, annotation and analysis of the genome sequence of the bacterial symbiont; based on this, it describes the metabolism of the symbiont and some potential roles within its host, such as the provision of precursors for cuticle synthesis via the shikimate pathway, and yet includes a phylogenetic analysis of EPSPS enzymes from different insect endosymbionts, including both, glyphosate sensitive and tolerant enzymes. Thus, the manuscript presents nice and interesting novel information. These considerations make the paper interesting for the readership of the Communications Biology journal.

But the abstract, the Ms is well written and structured, although it could be synthesized in some parts. Some details are missing in the materials and methods section.

Specific comments are the following:

Along the whole document: Latin names (taxa) should be written in italics, as well as gene names (some examples are given below)

Thank you for pointing this out. While not italicizing gene names was our oversight, according to the International Codes of Nomenclature only genus and species names are italicized, or the term *Candidatus* for proposed species names, but no other taxonomic levels.

Abstract: from my point of view, this is the poorest part of the article and it could be rewritten for clarity. For instance:

Lines 25-27 "decreased titers of the aromatic amino acid tyrosine in symbiont-depleted beetles support the ability to synthesize phenate..." ??? Maybe you mean "decreased titers of the aromatic amino acid tyrosine in symbiont-depleted beetles support the ability of symbionts to synthesize phenate..."

We changed the sentence to "decreased titers of the aromatic amino acid tyrosine in symbiont-depleted beetles support the symbionts' ability to synthesize prephenate"

Lines 28-29: "Glyphosate exposure inhibited symbiont establishment during development", which development??

Done – added “host”

Introduction:

Line 89: Bacteroidetes in italics

In accordance with nomenclature rules, we keep ‘Bacteroidetes’ in standard font.

Line 90: Coleoptera, Silvanidae in italics

In accordance with nomenclature rules, we keep ‘Coleoptera’ and ‘Silvanidae’ in standard font.

Results:

Lines 116-119: it should be explained better why you focus on the Bacteroidetes endosymbiont. Do you get sequences only from one bacterium within the bacteriome??

In our previous studies (Hirota et al. 2017, Engl et al. 2018) we identified monocultures of Bacteroidetes symbionts in the bacteriomes of *O. surinamensis*, using amplification of the phylogenetically informative 16S rRNA gene, cloning and Sanger sequencing of amplification products, as well as Fluorescence in situ hybridization.

Thus, we also focused on sequences classified as Bacteroidetes bacteria during the genome assembly.

Lines 124 and 125: when you write kbp, it should be bp

Done

Line 143: endosymbiont should be endosymbionts

Done

Line 158: this section title is a bit odd. Written like this it seems that *O. surinamensis* has just one symbiont (I am not sure if that is the case). In any case, why not to use already the proposed symbiont name? Something like: *Candidatus Shikimatogenerans silvanidophilus* OSUR encodes glycolysis and shikimate pathways. Just a suggestion...

Thank you for this suggestion. We changed the section title accordingly.

Lines 168 and 169: *Carsonella rudii* and *Nardonella* in italics

For proposed, yet unverified species names, only the term *Candidatus* is written in italics.

Line 195: *Shikimatogenerans silvanidophilus* in italics

For proposed, yet unverified species names, only the term *Candidatus* is written in italics.

Line 196: for “the symbiont” should be “for this symbiont”.

Done

Lines 200-202: I do not understand what you mean with this.

Beetles of the family Silvanidae are also called Silvanid beetles. To clarify we changed the sentence to: "Thus, we propose *silvanidophilus* as species name to indicate that this symbiont is associated with beetle of the family Silvanidae."

Lines 209-215: too much explanation for a results section.

Yes, these two sentences are taking up previous findings, but only to supply the reader with a reasoning and expectation why we chose to measure different amino acid titers, similar as requested by reviewer 3 for lines 179-184. We thus opt to retain the comment.

Line 243: the gene name (*aro*) should be in italics.

Done. Also for every other gene.

Lines 179-184: I miss a short explanation to the reason to do this.

We assume this comment refers to lines 279-284. Thank you for pointing this out. We added the following comment to clarify our motivation for this analysis:

"Finally, to assess how common symbiont mediated sensitivity to glyphosate is among insects engaging in nutritional symbioses, we tested whether the EPSPS of other intracellular symbionts besides the one of *S. silvanidophilus* belong to glyphosate sensitive EPSPS variants by a phylogenetic analysis of their amino acid sequences²⁵."

Discussion:

Lines 302: gene names should be in italics

Done

Materials and methods:

"Elimination of *O. surinamensis* symbionts" section: details such as the antibiotic concentration are missing.

The details can be found in the referenced publication (#42: Engl et al. 2018), but we added the focal point, the tetracycline concentration of 150mg / 5g oat flakes in lines 414/515.

"Symbiont genome sequence, assembly and annotation": it is not explained at all how you get X "bacteriomes dissected". I do not understand at all what a bacteriome dissection is, and, in any case, it is not explained how it was obtained.

Thank you for the question. By dissection we refer to the encyclopedic meaning of cutting open an animal to identify and separate single organs. In this context we removed the bacteriomes from the

rest of the beetle body and extracted DNA from only these organs, reducing the amount of host DNA to enrich resulting libraries in symbiont DNA and thereby facilitate the genome assembly. 24 bacteriomes result from six individual beetles, as each individual contains four bacteriomes.

We added a corresponding remark in L435: "(each individual beetle contains four separate bacteriomes)"

Line 463: name of the gene in italics

Done

"Comparison of bacteria": this section title is too broad. Please, choose a title representative of what you explain in this section.

We changed the section title to: "Comparison with other Bacteroidetes bacteria".

Supplementary Table 5 contains the most important information of the article, according to the way the work has been presented, since it shows how glyphosate impacts the beetle's development. Thus, I would say that it should not be a "supplementary file" and, in contrast, be included with the main text.

While we already report on the results of these statistical tests in a summarized manner in the results (L243-283) and Figure 5 we are happy to include the detailed results in the novel Table 3 instead of Supplementary Table 5. Following the same logic, we also moved Supplementary Table 4 into the main manuscript as Table 2 presenting test results on amino acid titers.

REVIEWERS' COMMENTS:

Reviewer #1 (Remarks to the Author):

Thank you for providing detailed responses to all concerns/questions. The most important of those (field relevance) has now been addressed appropriately, and I have no further concerns with this very interesting study.

Reviewer #2 (Remarks to the Author):

The revisions made by Dr. Engl and co-authors have addressed my previous comments. I believe the revised manuscript is acceptable for publication in Communications Biology and will be of interest to a wide audience.

Reviewer #3 (Remarks to the Author):

The revised version of the manuscript "Inhibition of a nutritional endosymbiont by glyphosate abolishes 2 mutualistic benefit on cuticle synthesis" by Kiefer has been improved. The manuscript provides important findings and conclusions are supported by the data. I believe the manuscript is ready for publication in its current form